

# Turbine- and farm-scale power losses in wind farms: an alternative to wake and farm blockage losses

Andrew Kirby[1], Takafumi Nishino[1], Luca Lanzilao[2], Thomas D. Dunstan[3], and Johan Meyers[2]

[1]Department of Engineering Science, University of Oxford, Parks Road, Oxford OX1 3PJ, UK
[2]Department of Mechanical Engineering, KU Leuven, Celestijnenlaan 300 – box 2421, B-3001 Leuven, Belgium
[3]Met Office, FitzRoy Road, Exeter EX1 3PB, UK

**Correspondence:** Andrew Kirby (andrew.kirby@trinity.ox.ac.uk)

**Abstract.** Turbine-wake and farm-atmosphere interactions can reduce wind farm power production. To model farm performance, it is important to understand the impact of different flow effects on the farm efficiency (i.e., farm power normalised by the power of the same number of isolated turbines). In this study we analyse the results of 43 large-eddy simulations (LES) of wind farms in a range of conventionally neutral boundary layers (CNBLs). First, we show that the farm efficiency $\eta_f$ is not well correlated with the wake efficiency $\eta_w$ (i.e., farm power normalised by the power of front row turbines). This suggests that existing metrics, classifying the loss of farm power into wake loss and farm blockage loss, are not best suited for understanding large wind farm performance. We then validate the assumption of scale separation in the two-scale momentum theory (Nishino & Dunstan, *J. Fluid Mech.*, vol. 894, 2020, p. A2) using the LES results. Building upon this theory, we propose two new metrics for wind farm performance, a turbine-scale efficiency $\eta_{TS}$, reflecting the losses due to turbine-wake interactions, and a farm-scale efficiency $\eta_{FS}$, indicating the losses due to farm-atmosphere interactions. The LES results show that $\eta_{TS}$ is insensitive to the atmospheric condition, whereas $\eta_{FS}$ is insensitive to the turbine layout. Finally, we show that a recently developed analytical wind farm model predicts $\eta_{FS}$ with an average error of 5.7% from the LES results.

## 1 Introduction

To meet future energy demands wind energy capacity will need to increase rapidly. It is likely that individual wind farms will become larger (Veers et al., 2022). When wind turbines are placed together in a farm they produce less power than in isolation. Predicting this power loss is key for designing wind farms. However, this remains difficult due to the multi-scale nature of wind farm aerodynamics (Porté-Agel et al., 2020).

Behind every turbine is a turbulent wake. When the wakes impact downstream turbines they can cause significant power losses. Turbine wakes have been investigated extensively using large-eddy simulations (LES) (e.g., Porté-Agel et al., 2013; Wu and Porté-Agel, 2015; Stevens et al., 2016) and wind tunnel experiments (e.g., Vermeer et al., 2003; Bastankhah and Porté-Agel, 2017; Hyvärinen et al., 2018). Measurements from wind farms show that downstream turbines produce less power





than the first upstream row (Barthelmie et al., 2010; Nygaard, 2014). Historically, this power degradation has been attributed to turbine-wake interactions.

Large wind farms can act as additional resistance to the atmospheric boundary layer (ABL) (Stevens and Meneveau, 2017). This can act to reduce the wind speed within and upstream of the farm (Porté-Agel et al., 2020). The upstream wind speed reduction is often referred to as the 'farm blockage' or 'global blockage' effect (Bleeg et al., 2018). LES of large wind farms (Wu and Porté-Agel, 2017; Allaerts and Meyers, 2017; Lanzilao and Meyers, 2024) show that an internal boundary layer forms in response to the increased flow resistance from the farm. The atmospheric response causes a velocity reduction within the

farm, in addition to the turbine wakes. How much of the downstream power degradation is caused by turbine wakes compared to the larger scale atmospheric response? It is important to make sure that we predict farm power losses by modelling the correct physical processes. Recently, Lanzilao and Meyers (2024) performed LES of large wind farms operating in conventionally neutral boundary layers (CNBLs), where the turbine layout and operating conditions were fixed but different ABL heights and thermal stratifications above the ABL were tested. Depending on these conditions of the atmosphere, the 'wake efficiency' $\eta_w$

(farm-averaged power normalised by the average power of the first row turbines) was found to vary significantly from 0.48 to 1.23. This raises the question: What physical processes are responsible for the different downstream power losses?

An alternative approach to understanding wind farm aerodynamics is the 'two-scale momentum theory' developed by Nishino and Dunstan (2020), who proposed to split the multi-scale problem into 'internal' turbine-scale and 'external' farm-scale sub-problems. The two sub-problems are coupled together by considering the conservation of momentum and matching

the farm-average wind speed. Using the two-scale momentum theory, Kirby et al. (2022) proposed the new concepts of turbine-scale and farm-scale power losses to understand farm performance, where the farm-average wind speed (rather than the wind speed upstream of the farm) plays a key role. The turbine-scale losses are due to farm-internal flow interactions (i.e., turbine-wake interactions), whereas the farm-scale losses are due to the atmospheric response to the whole farm (i.e., reduction of farm-average wind speed).

In this study we compare the two different classifications of wind farm power losses using LES of large finite-size wind farms. We use the LES results reported by Lanzilao and Meyers (2024) and also perform new simulations with different turbine layouts, which allow us to fully validate the 'two-scale separation' assumption and thus the concepts of turbine- and farm-scale losses. The LES data are available in a public database (Lanzilao and Meyers, 2023b). We first summarise the two-scale momentum theory in Sect. 2. The LES methodology is then briefly described in Sect. 3. A validation of the two-scale

separation assumption along with the turbine- and farm-scale losses are presented in Sect. 4. We also compare the farm-scale losses from the wind farm LES with predictions from an analytical wind farm model in Sect. 4. The results are discussed in Sect. 5 and concluding remarks given in Sect. 6.



## 2 Theory

### 2.1 Two-scale momentum theory

By considering the momentum balance for a control volume with and without a wind farm present, Nishino and Dunstan (2020) derived the non-dimensional farm momentum (NDFM) equation:

$$C_T^* \frac{\lambda}{C_{f0}} \beta^2 + \beta^\gamma = M \tag{1}$$

where $\beta$ is the farm wind-speed reduction factor which is defined as $\beta \equiv U_F/U_{F0}$ (where $U_F$ is the average wind speed in the nominal farm-layer of height $H_F$, and $U_{F0}$ is the farm-layer-averaged speed without turbines present); the (farm-averaged) 'internal' turbine thrust coefficient $C_T^*$ is defined as $C_T^* \equiv \sum_{i=1}^n T_i / \frac{1}{2}\rho U_F^2 nA$ (where $T_i$ is the thrust of turbine $i$, $n$ is the number of turbines in the farm and $A$ is the rotor swept area); the array density $\lambda$ is defined as $\lambda \equiv nA/S_F$ (where $S_F$ is the farm area); the natural surface friction coefficient $C_{f0}$ is defined as $C_{f0} \equiv \tau_{w0} / \frac{1}{2}\rho U_{F0}^2$ (where $\tau_{w0}$ is the bottom shear stress without turbines present); $\gamma$ is the bottom friction exponent defined as $\gamma \equiv \log_\beta(\tau_w/\tau_{w0})$ (assumed to be 2.0 in this study, following Nishino and Dunstan (2020) and also as justified later in Sect. 4.2); and $M$ is the momentum availability factor defined by $M \equiv M_F/M_{F0}$, where $M_F$ is the net momentum flux into the farm control volume with the turbines present and $M_{F0}$ the case without the turbines present. In this study we use a fixed definition of the farm-layer height $H_F = 2.5H_{hub}$ (where $H_{hub}$ is the turbine hub-height) following Kirby et al. (2022).

Patel et al. (2021) used numerical weather prediction (NWP) simulations to calculate $M$ for a realistic offshore wind farm site in the North Sea. They found, for most cases, an approximately linear relationship between $M$ and $\beta$. Therefore $M$ can be modelled as

$$M = 1 + \zeta(1 - \beta) \tag{2}$$

where $\zeta$ is called the wind extractability factor. Kirby et al. (2022) showed that $\zeta$ was a time-dependent parameter which varied with atmospheric conditions and inversely with farm size. More recently, Kirby et al. (2023b) proposed an analytical model of $\zeta$, as discussed later in Section 4.3.

Equations (1) and (2) can be solved to calculate the farm wind-speed reduction factor $\beta$ for a given farm design and atmospheric condition (i.e. $\lambda$, $C_T^*$, $C_{f0}$, $\gamma$ and $\zeta$). Using $\beta$, the farm power can be calculated using

$$C_p = \beta^3 C_p^* \tag{3}$$

where the (farm-averaged) turbine power coefficient is defined as $C_p \equiv \sum_{i=1}^n P_i / \frac{1}{2}\rho U_{F0}^3 nA$ ($P_i$ is power of turbine $i$ in the farm) and the (farm-averaged) 'internal' turbine power coefficient defined as $C_p^* \equiv \sum_{i=1}^n P_i / \frac{1}{2}\rho U_F^3 nA$.



## 2.2 Analytical model of ideal wind farm performance

Generally, the 'internal' turbine thrust coefficient $C_T^*$ depends on the turbine layout (Kirby et al., 2022). However, as suggested by Nishino (2016) and later confirmed by Kirby et al. (2022), an approximate upper limit of $C_T^*$ can be predicted by using an analogy to the classical actuator disc theory

$$C_T^* = 4\alpha(1-\alpha) = \frac{16C_T'}{(4+C_T')^2} \tag{4}$$

where $C_T' \equiv T_i/\frac{1}{2}\rho U_{T,i}^2 A$ is a turbine resistance coefficient which represents the turbine operating conditions, and $U_{T,i}$ is the streamwise velocity averaged across the rotor swept area of turbine $i$. Kirby et al. (2022) showed that some specific turbine layouts could exceed this $C_T^*$ value slightly, presumably due to local blockage effects (Nishino and Draper, 2015).

The two-scale momentum theory described in Sect. 2.1 can predict the performance of arrays of actuator discs (or aerodynamically ideal turbines operating below rated conditions). For actuator discs $C_p^* = \alpha C_T^*$ where $\alpha \equiv U_T/U_F$ is the local wind-speed reduction factor, and $\alpha$ can be estimated using $\alpha = \sqrt{C_T^*/C_T'}$. It is useful to note that this is strictly valid only for infinite regular arrays of actuator discs where the thrust of each turbine is identical to the farm-averaged turbine thrust. The (farm-averaged) power coefficient of an actuator disc is given by

$$C_p = \beta^3 \alpha C_T^* = \beta^3 C_T^{*\frac{3}{2}} C_T'^{-\frac{1}{2}}. \tag{5}$$

Using the analytical model of $C_T^*$ (Eq. (4)), Eq. (1), (2) and (5) can be solved to give a theoretical prediction of 'ideal' wind farm performance, denoted $C_{p,Nishino}$. If we assume $\gamma = 2.0$, we can derive a single analytical expression for $C_{p,Nishino}$, i.e.,

$$C_{p,Nishino} = \frac{64C_T'}{(4+C_T')^3} \times$$

$$\left[ \frac{-\zeta + \sqrt{\zeta^2 + 4\left(\frac{16C_T'}{(4+C_T')^2}\frac{\lambda}{C_{f0}}+1\right)(1+\zeta)}}{2\left(\frac{16C_T'}{(4+C_T')^2}\frac{\lambda}{C_{f0}}+1\right)} \right]^3. \tag{6}$$

Meanwhile, the power coefficient of an isolated turbine $C_{p,Betz}$ is given by

$$C_{p,Betz} = \frac{64C_T'}{(4+C_T')^3} \tag{7}$$

which gives a maximum turbine performance of $C_{p,Betz} = 16/27$ when $C_T' = 2.0$. These two theoretical predictions of performance ($C_{p,Nishino}$ and $C_{p,Betz}$) will be used to define the turbine-scale and farm-scale efficiencies later in Sect. 4.3.



# 3 Large-eddy simulation methodology

In this paper we analyse the LES of wind farms in CNBLs performed by Lanzilao and Meyers (2024) with 5 new simulation cases. Here we briefly summarise the main details of LES methodology, for more details see Lanzilao and Meyers (2024).

The turbines are modelled using an actuator disc model with no rotation (Calaf et al., 2010; Meyers and Meneveau, 2010).
The turbine forces are projected onto the numerical grid using a Gaussian convolution filter (Calaf et al., 2010). Recently, Shapiro et al. (2019) proposed an additional correction factor for actuator disk models to avoid over-prediction of power and thrust. Unfortunately, this correction factor was not yet included in the LES database of Lanzilao and Meyers (2024), and therefore it was also not used for the additional cases performed here. Instead, as a next best approximation, we use the correction factor of Shapiro et al. (2019) in a postprocessing step (see Sect. 4.3 for more details). The turbines have a diameter
$D$ of 198 m and a hub height $H_{hub}$ of 119 m. The thrust is calculated using a disk-based thrust coefficient of $C_T' = 1.94$ giving a traditional thrust coefficient of $C_T = 0.88$. A yaw controller is used to keep all turbine discs perpendicular to the incident flow to each turbine.

Table 1 summarises the wind farm designs considered in this study. In addition to the 'standard' design used by Lanzilao and Meyers (2024), we also consider 3 additional designs, namely 'aligned', 'half length' and 'double spacing'. The 'standard' farm
consists of 16 rows and 10 columns of turbines in a staggered layout. The streamwise and spanwise spacing between turbines is $5D$ giving a capacity density of approximately 10 MW km$^{-2}$. This is a dense wind farm but this density is being considered in some development areas. The farm has a length of 14.85 km and a width of 9.4 km. For the 3 additional farm designs (aligned, half length and double spacing) the turbine layout, farm length and turbine spacing were changed, respectively, from the standard design (table 1). Note that the farm length is the distance between the first and last turbine rows, and the 'half
length' case has 8 rows rather than 16 rows. For all simulations the computational domain size is $L_x \times L_y \times L_z = 50$ km$\times$ 30 km $\times$ 25 km. The grid resolution is $\Delta x = 31.25$ m, $\Delta y = 21.74$ m and $\Delta z = 5$ m in the lowest 1.5 km of the domain, following the set-up used by Lanzilao and Meyers (2024).

**Table 1.** A summary of wind farm designs considered in this study.

| Design | Turbine layout | Farm length | Farm width | Turbine spacing | Number of turbines |
|---|---|---|---|---|---|
| Standard | Staggered | 14.85 km | 9.4 km | $5D \times 5D$ | 160 |
| Aligned | Aligned | 14.85 km | 9.4 km | $5D \times 5D$ | 160 |
| Half length | Staggered | 6.93 km | 9.4 km | $5D \times 5D$ | 80 |
| Double spacing | Staggered | 14.85 km | 9.4 km | $10D \times 10D$ | 40 |

The bottom boundary conditions are given by the classical Monin-Obukhov similarity theory (Moeng, 1984). Periodic boundary conditions are applied at the streamwise and spanwise edges of the domain. The inflow condition is given by a
concurrent precursor simulation (Stevens et al., 2014) which is applied using a wave-free fringe technique that avoids spurious



excitation of gravity waves (Lanzilao and Meyers, 2023a). Moreover, to prevent the reflection of gravity waves, a Rayleigh damping layer is applied in the upper part of the domain (Lanzilao and Meyers, 2023a).

The atmospheric stratification is varied by changing the capping inversion height, capping inversion strength and free-atmosphere lapse rate. Table 2 shows a summary of the different atmospheric stratifications. All combinations of these pa-
rameters were considered for the 'standard' farm design by Lanzilao and Meyers (2024). We use the notation introduced by Lanzilao and Meyers (2024), e.g., H500-C5-G4 refers to a capping inversion height of 500 m, capping inversion strength of 5 K and free-atmosphere lapse rate of 4 K km$^{-1}$.

**Table 2.** A summary of atmospheric stratifications.

| | |
|---|---|
| Capping inversion height [m] | 1000, 500, 300, 150 |
| Capping inversion strength [K] | 2, 5, 8 |
| Free-atmosphere lapse rate [K km$^{-1}$] | 1, 4, 8 |

## 4    Results

In the following we first investigate the wake and farm-blockage losses observed in the LES performed by Lanzilao and
Meyers (2024) in Sect. 4.1. We then present in Sect. 4.2 a validation of the 'two-scale separation' assumption in the two-scale momentum theory proposed by Nishino and Dunstan (2020). Finally, we apply the concepts of 'turbine-scale' and 'farm-scale' losses (Kirby et al., 2022) to the LES results in Sect. 4.3.

### 4.1    Wake and farm blockage losses

Here we reanalyse the results of wind farm LES performed by Lanzilao and Meyers (2024), who reported that the farm
normalised power relative to the first-row power (i.e., wake efficiency) varied from 0.48 to 1.23 for the same turbine layout and wind direction. The aim of this section is therefore to investigate the physical mechanisms changing the farm normalised power.

There are 3 commonly used efficiencies for wind farm performance. Firstly, the 'wake efficiency' $\eta_w$ (sometimes called 'normalised power') is defined as

$$\eta_w \equiv \frac{P_{farm}}{P_1} \tag{8}$$

where $P_{farm}$ is the farm-averaged turbine power and $P_1$ is the first-row-averaged turbine power. Secondly, the 'non-local' efficiency $\eta_{nl}$ is defined by (Allaerts and Meyers, 2018)





$$\eta_{nl} \equiv \frac{P_1}{P_\infty} \tag{9}$$

where $P_\infty$ is the power output of an isolated turbine under the same atmospheric conditions. This represents the power loss due to the velocity reduction in front of the farm, i.e., due to farm blockage. Finally the 'farm efficiency' $\eta_f$ is defined as

$$\eta_f \equiv \frac{P_{farm}}{P_\infty} \equiv \eta_w \eta_{nl}. \tag{10}$$

The farm efficiency $\eta_f$ quantifies the overall power losses caused by placing turbines together in a farm.

As noted by Lanzilao and Meyers (2024), the farm LES results show a relatively strong negative correlation between $\eta_w$ and $\eta_{nl}$ (figure 1). When the farm blockage increases (i.e. $\eta_{nl}$ decreases), the downstream power losses decrease (i.e. $\eta_w$ increases). These two effects counteract each other to a certain extent. This means that $\eta_w$ is affected not only by turbine-wake interactions but also by larger farm-scale flow effects causing farm blockage.

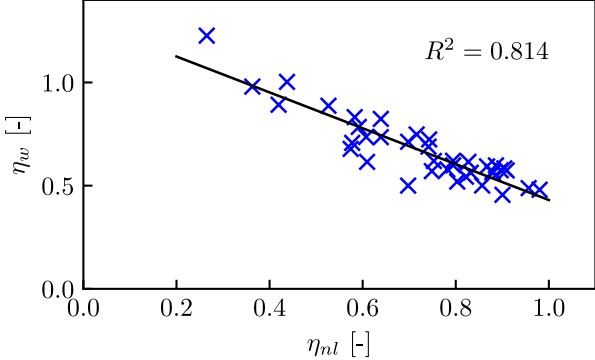

**Figure 1.** Relationship between wake efficiency $\eta_w$ and non-local efficiency $\eta_{nl}$ for all 38 LES cases from Lanzilao and Meyers (2024). The $R^2$ value shows the coefficient of determination.

The correlation between $\eta_w$ and $\eta_{nl}$ is caused by the induced pressure gradients across the farm. To illustrate this, the pressure perturbation for cases H300-C2-G1 and H300-C8-G1 is shown in Fig. 2. Case H300-C2-G1 has a low degree of farm blockage ($\eta_{nl} = 0.857$) and a relatively small induced pressure gradient. Conversely, H300-C8-G1 has a high degree of farm blockage ($\eta_{nl} = 0.437$) and a large induced pressure gradient. Essentially, H300-C8-G1 has a larger driving force across the farm, meaning that the velocity tends to stay high across the farm despite the lower velocity at the front. This causes $\eta_w$ to be higher ($\eta_w = 1.00$ compared to $\eta_w = 0.501$ for H300-C2-G1).

The LES results also show that the overall farm efficiency $\eta_f$ is not well correlated with either the wake efficiency $\eta_w$ or the non-local efficiency $\eta_{nl}$. Figure 3(a) shows the weak correlation between $\eta_f$ and $\eta_w$. This shows that the 'wake efficiency' or 'normalised power' is not a good indicator of wind farm efficiency. The correlation between $\eta_f$ and $\eta_{nl}$ is also relatively weak

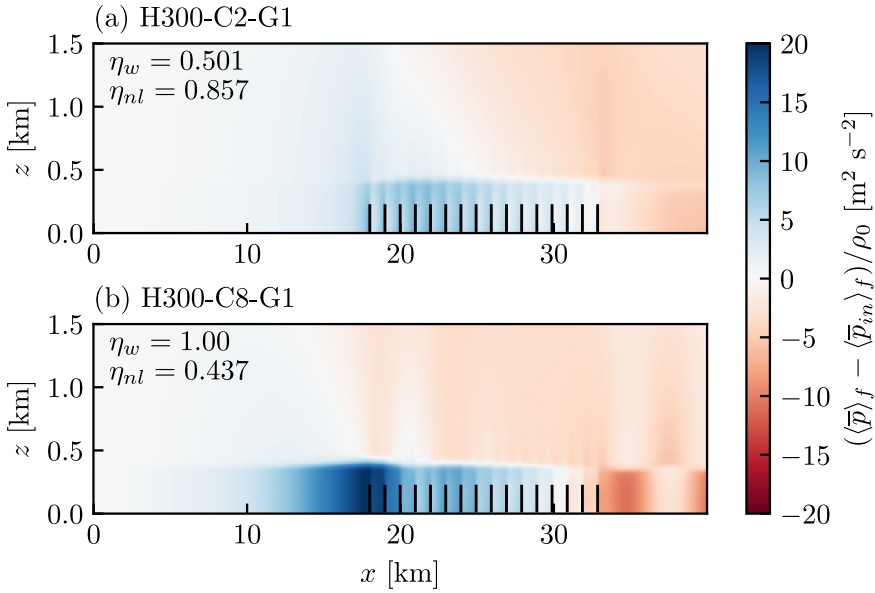

**Figure 2.** Time-averaged pressure perturbation averaged across the farm width for cases **(a)** H300-C2-G1 and **(b)** H300-C8-G1.

as shown in Fig. 3(b). The negative farm blockage effect is mostly counteracted by the increased pressure gradient across the farm which increases $\eta_w$.

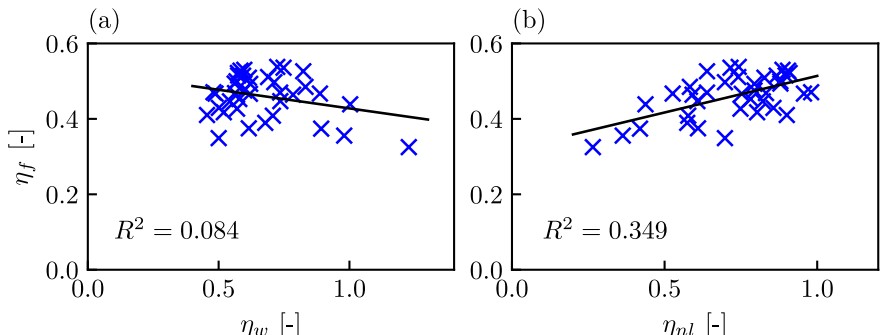

**Figure 3. (a)** Relationship between farm efficiency $\eta_f$ and wake efficiency $\eta_w$ and **(b)** relationship between farm efficiency $\eta_f$ and non-local efficiency $\eta_{nl}$, for all 38 LES cases from Lanzilao and Meyers (2024). The $R^2$ values shows the coefficient of determination.

To better understand why the same turbine layout results in a very wide range of $\eta_w$ from 0.48 to 1.23, now we will investigate whether this could be explained by either 1) different 'effective' turbine layouts caused by changes in local wind directions within the farm, or 2) different wake recovery rates. In the following, we will again focus on the two illustrative






cases, H300-C2-G1 and H300-C8-G1. The capping inversion height is the same for both but H300-C2-G1 gives $\eta_w = 0.501$ whereas H300-C8-G1 gives $\eta_w = 1.00$.

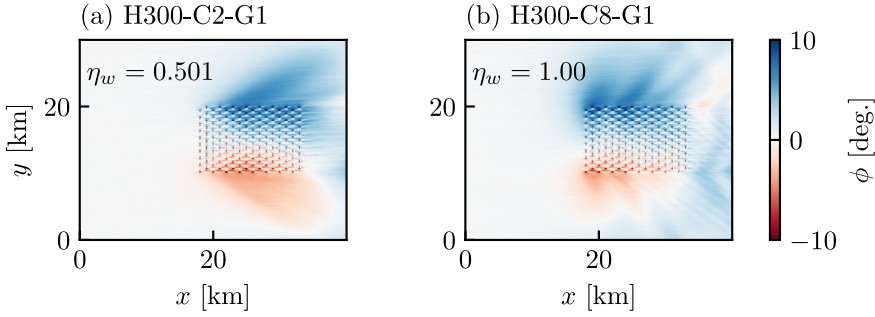

**Figure 4.** Time-averaged flow angle at the turbine hub height for cases **(a)** H300-C2-G1 and **(b)** H300-C8-G1.

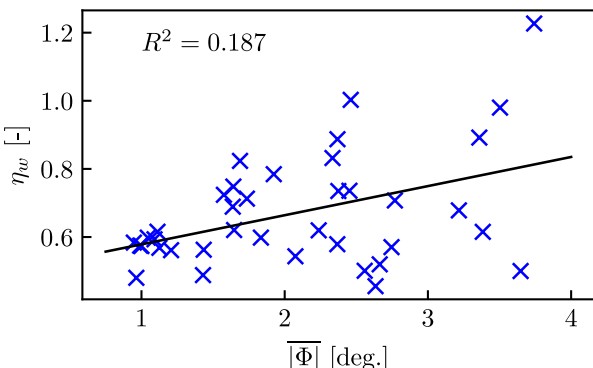

**Figure 5.** Relationship between wake efficiency $\eta_w$ and the farm-averaged absolute turbine yaw angle $\overline{|\Phi|}$ for all 38 LES cases from Lanzilao and Meyers (2024). The $R^2$ value shows the coefficient of determination.

First, we show that the large difference in $\eta_w$ cannot be explained by different 'effective' turbine layouts. The local flow direction for cases H300-C2-G1 and H300-C8-G1 is shown in Fig. 4. Both cases have an outward flow direction of approxi-
mately $5^o$ at the sides. However, both cases have similar variations of the local flow directions across the farm. Figure 5 shows there is not a strong relationship between $\eta_w$ and the farm-averaged absolute turbine yaw angle $\overline{|\Phi|}$ for all 38 cases, indicating that the large variation of $\eta_w$ cannot be explained by the change of effective turbine layout.

Next, we show that the wake recovery behind each turbine is also uncorrelated with the wake efficiency $\eta_w$. The farm flow profiles are shown in Fig. 6(a) for H300-C2-G1 and in Fig. 6(c) for H300-C8-G1. The individual wake deficits look similar but
rather it is the farm-scale flows which are different. A closer view of individual wakes towards the center of the farm is shown





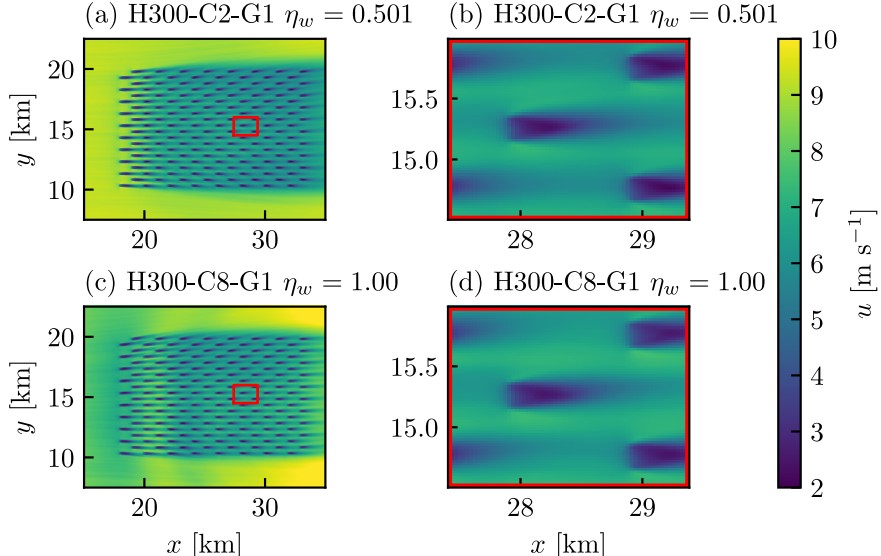

**Figure 6.** Time-averaged $u$ velocity contours at the turbine hub height for case H300-C2-G1 **(a)** across the whole farm and **(b)** inside the farm; and for case H300-C8-G1 **(c)** across the whole farm and **(d)** inside the farm.

in Fig. 6(b) and 6(d). Despite one case having $\eta_w = 0.501$ and the other having $\eta_w = 1.00$, the wakes look almost identical. This suggests that the characteristics of individual turbine wake recovery are not contributing to the difference in $\eta_w$.

A more quantitative comparison in Fig. 7 shows that both cases have similar wake velocity deficits. Here we calculated the wake velocity deficit by defining new coordinates $x_i$ and $y_i$ local to each turbine (see Fig. 7(b)). $x_i$ is perpendicular to
each rotor and $y_i$ parallel. We averaged the wake velocity deficits (relative to the undisturbed velocity $u_\infty(z)$ recorded in the precursor simulation) for each turbine in the 11th row. This row was chosen because the flow profiles are characteristic of the average flow profile across the entire farm. Figures 7(c)-(f) show that both horizontal and vertical wake deficit profiles are approximately Gaussian, and the wake deficit profiles and wake recovery rate are both similar. H300-C8-G1 has a slightly smaller normalised wake velocity deficit compared to H300-C2-G1, but this is not sufficient to explain the large difference in
$\eta_w$ values.

The wake recovery across the entire farm is also similar for the two cases. We calculated the wake width by fitting a Gaussian function to the wake deficit profiles shown in Fig. 7. Note that the centre of Gaussian function was not fixed to the rotor centre. The wake width $\sigma$ was calculated as the geometric mean of the wake width in the horizontal $\sigma_y$ and vertical $\sigma_z$ directions, i.e., $\sigma = \sqrt{\sigma_y \sigma_z}$. The wake width as a function of downstream distance for turbine rows 3 to 16 is shown in Fig. 8. The first two
rows were excluded as the wake recovery was much slower. An approximately linear growth in wake width can be seen for both cases. We calculated the wake expansion coefficient $k^*$ using the equation $\sigma/D = k^* x_i/D + \epsilon$ where $\epsilon$ is the initial wake width. We then averaged the value of $k^*$ across the 3rd to 16th rows to obtain a farm-averaged $k^*$, which is shown in Fig. 8. The value of $k^*$ was found to be higher than the values reported by Bastankhah and Porté-Agel (2014). This is presumably

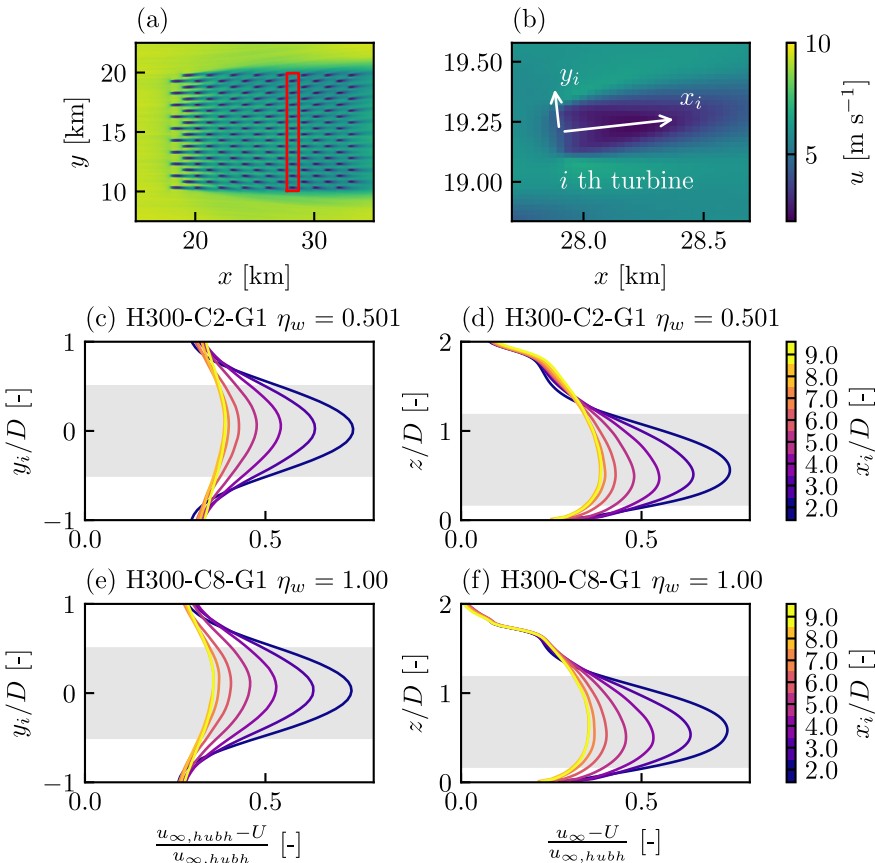

**Figure 7. (a)** Time-averaged $u$ velocity contours at the turbine hub height for case H300-C2-G1 with the 11th row highlighted in red, and **(b)** new 'local' coordinates $x_i$ and $y_i$. Normalised wake velocity deficit profiles averaged for all 10 turbines in the 11th row, plotted in the horizontal ($y_i$) direction at the hub height and in the vertical ($z$) directions through the rotor center, respectively, for **(c)**, **(d)** case H300-C2-G1 and **(e)**, **(f)** case H300-C8-G1.

because the turbulence levels are higher within a large wind farm. Most importantly, the average wake growth rate is higher for the case with the lower value of $\eta_w$. This again demonstrates that $\eta_w$ is not strongly related to local wake recovery behind each turbine.

To confirm this trend further, we calculated the farm-averaged wake expansion coefficient $k^*$ for 29 of the farm LES cases from Lanzilao and Meyers (2024). The cases with the lowest capping inversion (150 m) were excluded as they did not have Gaussian wake deficit profiles in the vertical direction, due to the vicinity of the capping-inversion base to the turbine-tip height. Figure 9(a) shows that there is no correlation between the farm efficiency $\eta_f$ and the wake expansion coefficient $k^*$. Figure 9(b) shows that low $\eta_w$ values cannot be explained by a slower wake recovery. On the contrary, these LES results show that cases with a low $\eta_w$ value tend to have a faster wake recovery.





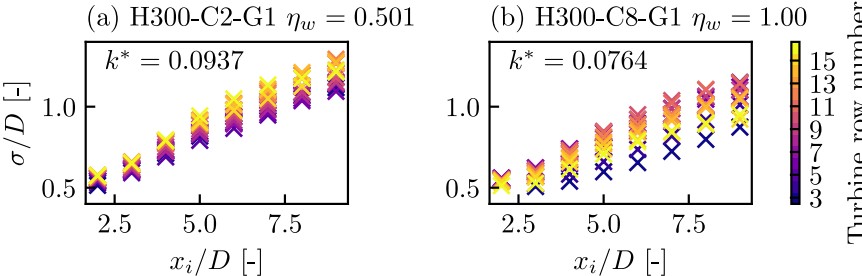

**Figure 8.** Normalised wake width with streamwise distance for different turbine rows for cases **(a)** H300-C2-G1 and **(b)** H300-C8-G1.

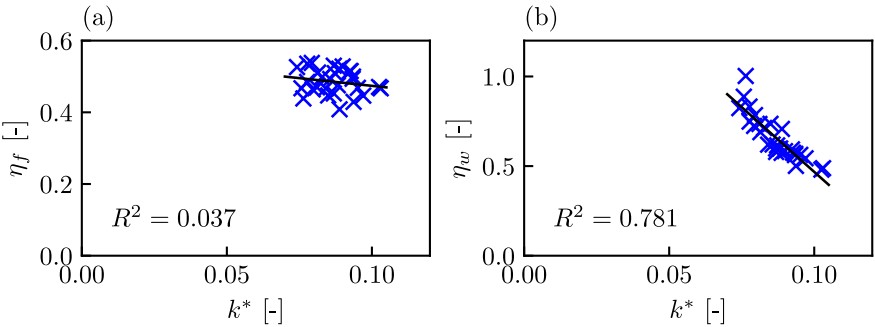

**Figure 9.** **(a)** Relationship between farm efficiency $\eta_f$ and wake expansion coefficient $k^*$ and **(b)** relationship between wake efficiency $\eta_w$ and wake expansion coefficient $k^*$. $R^2$ shows the coefficient of determination.

The wake efficiency $\eta_w$ has been extensively used to analyse farm performance as it is a relatively easy parameter to calculate, e.g. using SCADA data. However, these LES results (for a fixed staggered turbine layout with different ABL conditions) suggest that this wake efficiency parameter $\eta_w$ is not a good indicator of the turbine-wake interactions within the farm.

### 4.2 Validation of two-scale separation assumption

The two-scale momentum theory provides an alternative way of understanding wind farm performance. This theory is particularly useful when the 'two-scale separation' assumption is valid, meaning that the farm 'internal' parameters ($C_T^*$ and $\gamma$) depend only on 'internal' or turbine-scale conditions, whereas the 'external' parameter ($\zeta$) depends only on 'external' or farm-scale conditions. This assumption allows the turbine-scale and farm-scale flows to be modelled separately; however, this assumption has not been fully validated in previous studies. In the following we present a first validation of the two-scale separation assumption using 4 new LES results as well as the previous LES results from Lanzilao and Meyers (2024).

Here we calculate $\zeta_{LES}$ directly from the momentum availability factor $M_{LES}$ and the farm wind-speed reduction factor $\beta_{LES}$ obtained from LES (Eq. (11)). $M_{LES}$ is given by Eq. (12a) which can be simplified into Eq. (12b) and (12c). The bottom





friction exponent $\gamma$ was not recorded in the present LES results, but we can expect that this varies between 1.5 and 2.0 (Kirby et al., 2022). Considering Eq. (12c), a typical value of $\lambda C_T^*/C_{f0}$ is 17.5 for the farms in this study meaning that the total force due to turbine thrust is much larger than the force due to the bottom friction (and hence, the impact of $\gamma$ is very small). For example, if we suppose that $\beta = 0.75$, using $\gamma = 1.5$ gives $M_{LES} = 10.5$ whereas $\gamma = 2.0$ gives $M_{LES} = 10.4$. Since the value of $M_{LES}$ is largely insensitive to the value of $\gamma$, we will use $\gamma = 2.0$ in the following analysis.

$$\zeta_{LES} = \frac{M_{LES} - 1}{1 - \beta_{LES}} \tag{11}$$

$$M_{LES} = \frac{\sum_{i=1}^{n} T_i + \tau_w S}{\tau_{w0} S} \tag{12a}$$

$$= \frac{\sum_{i=1}^{n} T_i}{\tau_{w0} S} + \beta_{LES}{}^{\gamma} \tag{12b}$$

$$= \frac{\lambda C_{T,LES}^*}{C_{f0}} \beta_{LES}{}^2 + \beta_{LES}{}^{\gamma} \tag{12c}$$

Figure 10 shows the relationship between $M_{LES}$ and $\beta_{LES}$ obtained for 3 different atmospheric conditions. As can be seen from the figure, the wind extractability factor $\zeta_{LES}$ changes with the atmospheric conditions but it is not sensitive to the turbine layout. The aligned turbine layouts result in a lower wind speed reduction (i.e., lower value of $1 - \beta_{LES}$) because they present a lower flow resistance. However, the value of $\zeta_{LES}$ is almost identical for aligned and staggered layouts under a given atmospheric condition. A farm with a staggered layout but doubled turbine spacing ($10D \times 10D$) also follows approximately the same relationship (see Fig. 10(b)). This demonstrates that the linear relationship is valid for a wide range of $\beta$. It can also be seen that $\zeta_{LES}$ decreases with decreasing capping inversion height. This trend was predicted by the theoretical model of $\zeta$ proposed by Kirby et al. (2023b). The values of $\zeta_{LES}$ for all atmospheric conditions tested in this study are shown in Fig. 11(a). As shown theoretically by Nishino and Dunstan (2020), for a given wind farm, there is a positive monotonic relationship between $\zeta$ and wind farm efficiency. Therefore effects of different atmospheric conditions on the wind farm efficiency reported by Lanzilao and Meyers (2024) can be explained by different wind extractability factors $\zeta$.

The LES results also show that the internal turbine thrust coefficient $C_{T,LES}^*$ is insensitive to atmospheric conditions (Fig. 11(b)). Apart from the lowest capping inversion cases (H150), the staggered turbine layout consistently gives a high $C_{T,LES}^*$ value of about 1.0 irrespective of atmospheric stratification (Fig. 11(b)), whereas the aligned turbine layout consistently gives a lower $C_{T,LES}^*$ value than the staggered one. This trend is expected as Kirby et al. (2022) showed that increased turbine-wake interactions reduce the value of $C_T^*$. Turbine-wake interactions reduce $C_T^*$ because the waked turbines experience a lower incident wind speed and so produce less thrust.

These LES results in Fig. 10 and 11 strongly indicate that the assumption of 'two-scale separation' is valid for large finite wind farms, at least in a practical range of CNBLs tested in this study. This means that the impact of turbine-scale flows (i.e., turbine-wake interactions) and farm-scale flows (i.e., farm-atmosphere interaction) could be modelled separately through





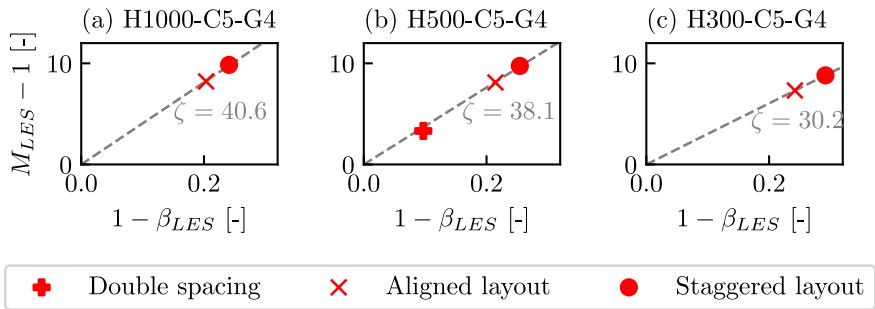

**Figure 10.** Relationship between momentum availability factor $M_{LES}$ and farm wind-speed reduction factor $\beta_{LES}$ for **(a)** H1000-C5-G4, **(b)** H500-C5-G4 and **(c)** H300-C5-G4 atmospheric conditions.

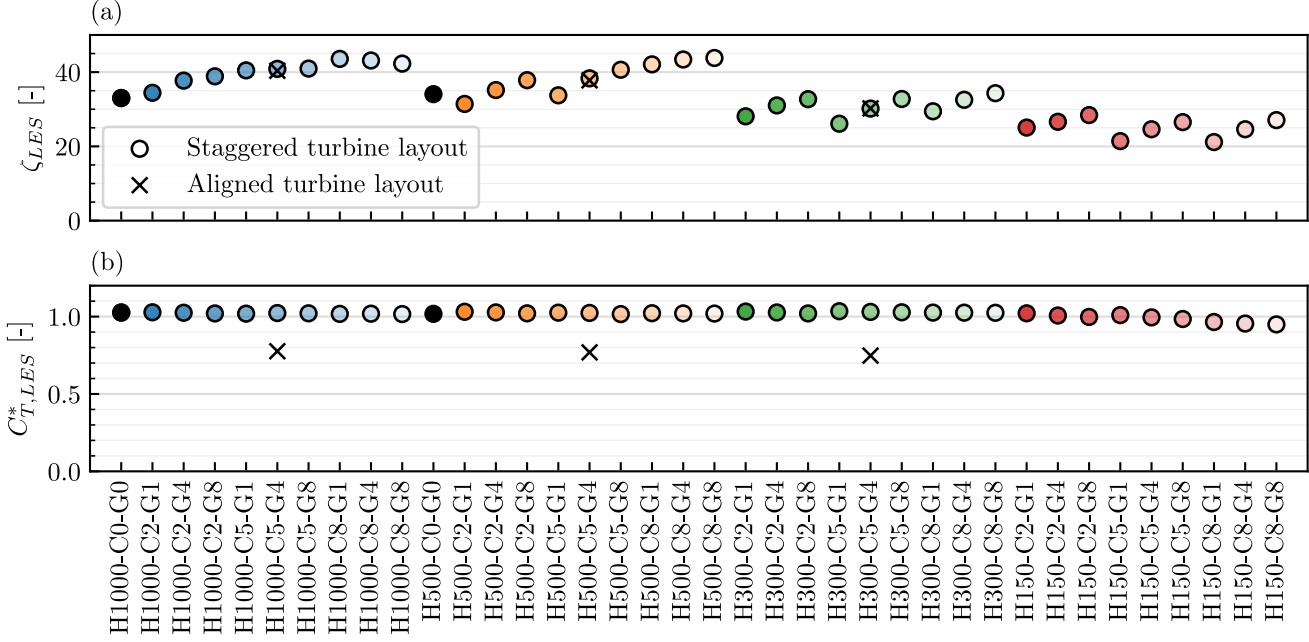

**Figure 11.** Values of **(a)** wind extractability factor $\zeta$ and **(b)** 'internal' turbine thrust coefficient $C_T^*$ for all atmospheric conditions tested.

the modelling of $C_T^*$ and $\zeta$, as suggested originally by Nishino and Dunstan (2020), to predict wind farm power in a less complicated and more physically meaningful manner.

### 4.3 Turbine-scale and farm-scale power losses

Turbine-scale and farm-scale power losses (Kirby et al. (2022); see also Stevens (2023)) are alternative metrics for wind farm performance, as illustrated in Fig. 12. The turbine-scale power losses are due to the internal flow interactions within the farm





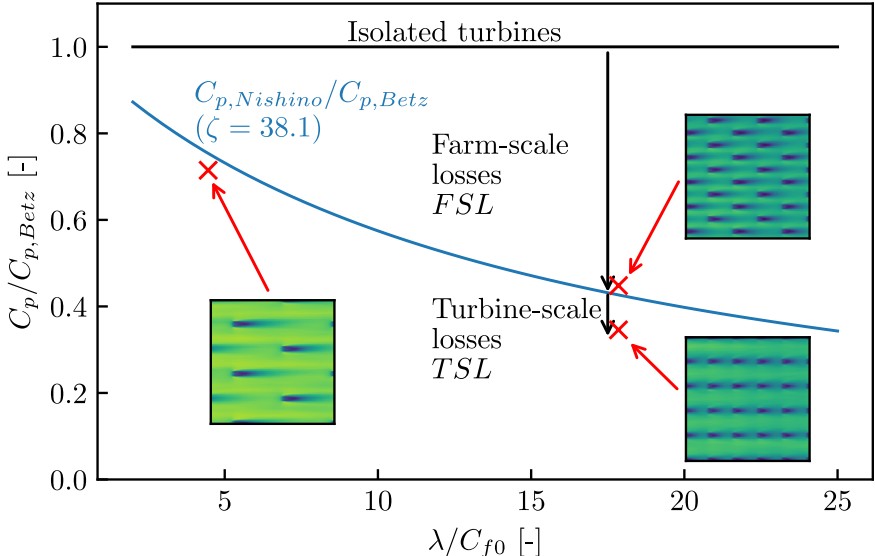

**Figure 12.** Schematic of the overall farm efficiency $C_p/C_{p,Betz}$ against the effective array density $\lambda/C_{f0}$, illustrating farm-scale and turbine-scale power losses. The blue line shows the ideal farm performance predicted by the two-scale momentum theory for a given set of conditions (corresponding to Fig. 10(b) with $\zeta = 38.1$), whereas the red crosses show the results of 3 farm LES cases discussed in Fig. 10(b).

(i.e., turbine-wake interactions). Farm-scale power losses are due to the interaction between the ABL and the farm as a whole. Here we propose a slight modification to these new metrics for wind farm performance, namely the 'turbine-scale efficiency' $\eta_{TS}$ and 'farm-scale efficiency' $\eta_{FS}$[1] defined as:

$$\eta_{TS} \equiv \frac{C_p}{C_{p,Nishino}} \tag{13}$$

$$\eta_{FS} \equiv \frac{C_{p,Nishino}}{C_{p,Betz}}. \tag{14}$$

$\eta_{TS}$ and $\eta_{FS}$ are related to the overall wind farm efficiency $C_p/C_{p,Betz}$ by

$$\frac{C_p}{C_{p,Betz}} \equiv \eta_{TS}\eta_{FS}. \tag{15}$$

Note that $C_p/C_{p,Betz}$ is slightly different from the overall farm efficiency $\eta_f$ used by Lanzilao and Meyers (2024). Here we normalise by the turbine performance predicted using the actuator disc theory, $C_{p,Betz}$, for a given turbine resistance coefficient

---

[1]Note that the turbine-scale efficiency $\eta_{TS}$ and farm-scale efficiency $\eta_{FS}$ are related to the 'turbine-scale loss factor' $\Pi_T$ and 'farm-scale loss factor' $\Pi_F$ introduced by Kirby et al. (2022) by the following expressions $\eta_{TS} \equiv 1 - \Pi_T$ and $\eta_{FS} \equiv 1 - \Pi_F$.



$(C_T' = 1.94$ in this study). This is instead of the isolated turbine power found using LES, $P_\infty$, which is slightly different from the power predicted by the actuator disc theory. We normalised all values by $C_{p,Betz}$ to ensure the predicted power coefficients

from the LES, $C_p$, and the theory, $C_{p,Nishino}$, are normalised by the same value. A summary of the efficiency metrics used in this study is given in table 3.

As can be seen from Eq. (15), the overall farm efficiency is the product of turbine-scale efficiency $\eta_{TS}$ and farm-scale efficiency $\eta_{FS}$. For convenience, we can also introduce an alternative set of metrics, namely the 'turbine-scale loss' $TSL$ defined in Eq. (16) and 'farm-scale loss' $FSL$ defined in Eq. (17). The only difference from $\eta_{TS}$ and $\eta_{FS}$ is that $TSL$ and

$FSL$ both have the same denominator $C_{p,Betz}$. This allows the two losses to be simply added up (instead of multiplied) to obtain the total loss as in Eq. (18).

$$TSL \equiv \frac{C_{p,Nishino} - C_p}{C_{p,Betz}} \equiv \eta_{FS}\left(1 - \eta_{TS}\right) \tag{16}$$

$$FSL \equiv \frac{C_{p,Betz} - C_{p,Nishino}}{C_{p,Betz}} \equiv 1 - \eta_{FS} \tag{17}$$

$$\frac{C_p}{C_{p,Betz}} \equiv 1 - (TSL + FSL) \tag{18}$$

**Table 3.** A summary of wind farm efficiency metrics.

| Efficiency metric | Notes |
|---|---|
| $\eta_f = P_{farm}/P_\infty$ | $P_{farm}$ - farm-averaged turbine power |
| | $P_\infty$ - isolated turbine power |
| $\eta_w = P_{farm}/P_1$ | $P_1$ - front-row-averaged turbine power |
| $\eta_{nl} = P_1/P_\infty$ | |
| $C_p/C_{p,Betz}$ | $C_p$ - farm-averaged power coefficient |
| | $C_{p,Betz}$ - ideal power coefficient for isolated turbines |
| $\eta_{TS} = C_p/C_{p,Nishino}$ | $C_{p,Nishino}$ - ideal power coefficient for turbines in a farm |
| $\eta_{FS} = C_{p,Nishino}/C_{p,Betz}$ | |

In this study we calculate $\eta_{TS}$ and $\eta_{FS}$ using $\zeta_{LES}$ obtained from each LES case. Shapiro et al. (2019) proposed a correction factor $N$ for the overpredicted velocity through an actuator disc as

$$N = \left(1 + \frac{C_T'}{2} \frac{1}{\sqrt{3\pi}} \frac{\Delta}{D}\right)^{-1}. \tag{19}$$





where $\Delta$ is the Gaussian kernel width used for projecting turbine forces onto the numerical grid. In this study $\Delta = 32.61$ m in the y and z directions, which gives $N = 0.950$, meaning that the turbine thrust and power are corrected by $N^2$ and

$N^3$, respectively. Since the correction factor of Shapiro et al. (2019) was not implemented in the LES, we apply it here as a postprocessing step, to correct for possible overpredictions of power and thrust.

To calculate $\eta_{TS}$ and $\eta_{FS}$ we used the procedure summarised in Fig. 13. Essentially, we solved Eq. (1) and (2) for $\beta$ using $\zeta = \zeta_{LES}$ and the parameter values in table 4. Note that we used $0.974$ as the value of $C_T^*$ which has been corrected (i.e., adjusted upwards) to account for LES resolution effects. The $5D \times 5D$ turbine spacing gives an array density $\lambda$ of $0.0314$. The

value of the farm-layer-height is given by $H_F = 2.5 H_{hub}$ and for the turbines simulated $H_{hub}$ is 119 m. $C_{p,Nishino}/C_{p,Betz}$ is given by the value of $\beta^3$.

Step 1.   Obtain $M_{LES}$ from $M_{LES} = \frac{\sum_{i=1}^{n} T_{i,LES}}{\tau_{w0} S} + \beta_{LES}^2$

Step 2.   Obtain $\zeta_{LES}$ from $\zeta_{LES} = \frac{M_{LES} - 1}{1 - \beta_{LES}}$

Step 3.   Obtain $\beta$ from $C_T^* \frac{\lambda}{C_{f0}} \beta^2 + \beta^2 = 1 + \zeta_{LES}(1 - \beta)$

n.b. $C_T^*$ in Eq. (4) is multiplied by $1/N^2$ to account for LES resolution effects

Step 4.   Obtain $\eta_{FS}$ from $\eta_{FS} = \beta^3$

Step 5.   Obtain $\eta_{TS}$ from $\eta_{TS} = \frac{1}{\beta^3} \times \frac{C_{p,LES}}{C_{p,Betz}}$

n.b. $C_{p,Betz}$ in Eq. (7) is multiplied by $1/N^3$ to account for LES resolution effects

**Figure 13.** Procedure used to calculate turbine-scale efficiency $\eta_{TS}$ and farm-scale efficiency $\eta_{FS}$.

Figure 14 compares the farm performance for 3 different atmospheric conditions (including the 2 cases discussed earlier in Sect. 4.1) for demonstration. As can be seen from figure 14(a), the wake and non-local efficiencies are both sensitive to capping inversion strength. These two effects mostly canceled each other out, giving similar farm efficiencies for these 3 cases.

Conversely, Fig. 14(b) shows that the turbine-scale efficiency $\eta_{TS}$ is almost unchanged for the 3 cases. This reflects the fact that the turbine layout was unchanged, so the turbine-scale flows were very similar (as shown earlier in Fig. 6). $\eta_{TS}$ is slightly greater than 1, which means that, on the turbine-scale (i.e., for a given farm-averaged wind speed) these 'clustered' turbines





**Table 4.** Parameter values used to calculate turbine-scale efficiency $\eta_{TS}$ and farm-scale efficiency $\eta_{FS}$.

| Quantity | Value |
|----------|-------|
| $C_T^*$ | 0.974 |
| $\lambda$ | 0.0314 |
| $H_F$ | 297.5 m |
| $\gamma$ | 2.0 |

perform slightly better than isolated turbines (this will be further discussed later in this section). The close agreement between $C_p/C_{p,Betz}$ and $\eta_{FS}$ means that all the power losses in these 3 cases are on the farm-scale, i.e., due to the farm-atmosphere

interaction.

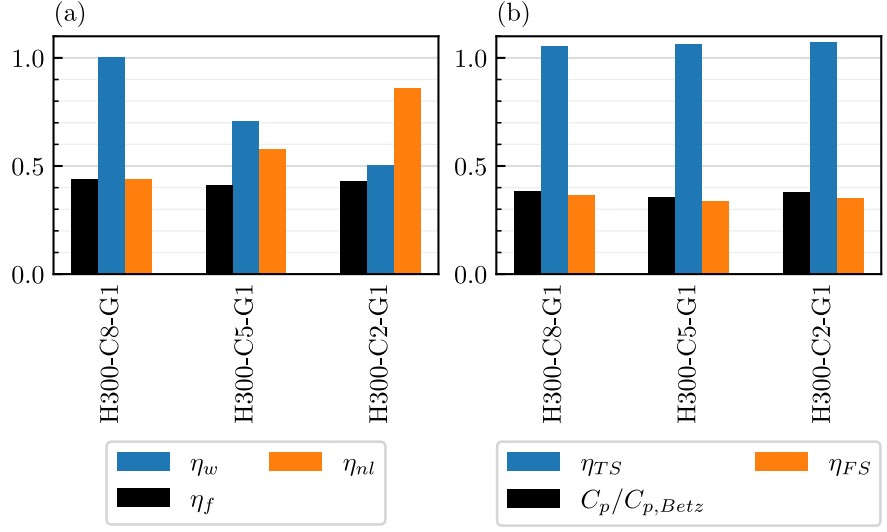

**Figure 14.** Comparison of wind farm performance for cases H300-C8-G1, H300-C5-G1 and H300-C2-G1 using **(a)** wake efficiency $\eta_w$ and non-local efficiency $\eta_{nl}$ and **(b)** turbine-scale efficiency $\eta_{TS}$ and farm-scale efficiency $\eta_{FS}$.

   Figure 15 shows the values of $\eta_{TS}$ and $\eta_{FS}$ for the same staggered farm under 38 different atmospheric conditions. Almost all the variation in $C_p/C_{p,Betz}$ is explained by $\eta_{FS}$. This reflects the physical observation that different stratifications affect the large-scale farm-atmosphere interaction, changing the power generation efficiency on the farm-scale. The value of $\eta_{TS}$ is nearly the same for most cases with a value of approximately 1.05 (discussed later in this section). The 6 cases with a lower

$\eta_{TS}$ are all with the lowest capping inversion height of $150$ m. These cases show a larger change in wind direction within the farm, changing the turbine-scale flow characteristics and thus $\eta_{TS}$.



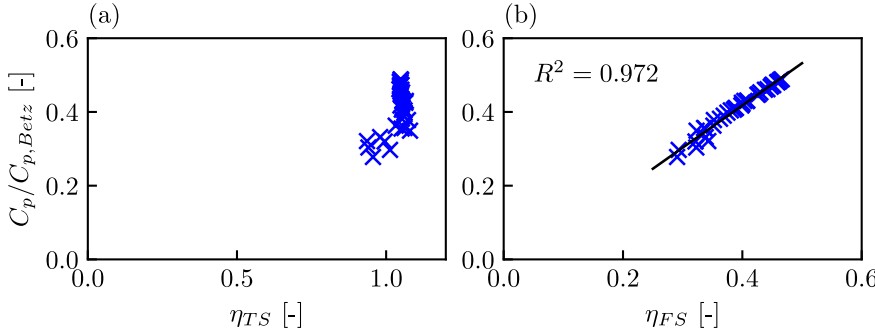

**Figure 15. (a)** Relationship between the overall farm efficiency $C_p/C_{p,Betz}$ and turbine-scale efficiency $\eta_{TS}$ and **(b)** farm-scale efficiency $\eta_{FS}$, for all 38 LES cases with the dense staggered turbine layout. The $R^2$ value shows the coefficient of determination.

Next, we examine how the impact of changing the turbine layout is captured by the new efficiency metrics, for the 3 atmospheric conditions discussed earlier in Fig. 10. As can be seen from Fig. 16, changing the layout from 'staggered' to 'aligned' changes the value of $\eta_{TS}$ from approximately 1.05 to just above 0.8 for all 3 cases. Given that in 'aligned' cases

the second row of turbines produces much less power than the first, it might appear that $\eta_{TS} \approx 0.8$ is fairly high. However, this is reasonable since the overall farm efficiency $\eta_f$ decreases by about 23% when the layout is changed from staggered to aligned for all 3 atmospheric conditions considered. The farm-scale efficiency $\eta_{FS}$ is practically unchanged with turbine layout. Conversely, the the existing metrics $\eta_w$ and $\eta_{nl}$ are both affected by the turbine layout, implying that the effect of turbine layout needs to be considered explicitly in the modelling of both wake and farm blockage losses. Using $\eta_{TS}$ and $\eta_{FS}$

instead allows us to separate the effect of turbine layout from the effect of atmospheric conditions.

It is also worth noting that, for all these cases, the power losses are larger on the farm-scale than on the turbine-scale. Figure 16 shows that $\eta_{FS}$ is smaller than $\eta_{TS}$ for both staggered and aligned layouts, despite the small turbine spacing ($5D$) considered in these cases. This agrees qualitatively with the predictions made by Kirby et al. (2022).

The new efficiency metrics $\eta_{TS}$ and $\eta_{FS}$ are also applicable to smaller farms and larger turbine spacings. Here we simulated

two additional layouts under the H500-C5-G4 atmospheric condition. In one case, 'half length', the streamwise length of the farm was halved to 6.93 km (shown in Fig. 17(b)). In the other case, 'double spacing', the turbine spacing was doubled in the $x$ and $y$ directions to $10D \times 10D$ (shown in Fig. 17(c)) whilst the farm size was kept constant. The results are compared with the 'standard' case in Fig. 18, showing that the changes in overall farm efficiency are mostly due to changes in $\eta_{FS}$. The 'half length' case gives a higher $\eta_{FS}$ because the farm-scale wind speed reduction (due to farm-atmosphere interaction) is less severe

for smaller wind farms. The 'double spacing' case gives an even higher $\eta_{FS}$ because of the low array density, which reduces the total farm thrust and thus the farm-atmosphere interaction compared to the 'standard' case. The turbine-scale efficiency $\eta_{TS}$ is similar and close to 1 for all 3 cases, reflecting the fact that turbine-wake interactions have limited impact for these staggered turbine arrays.





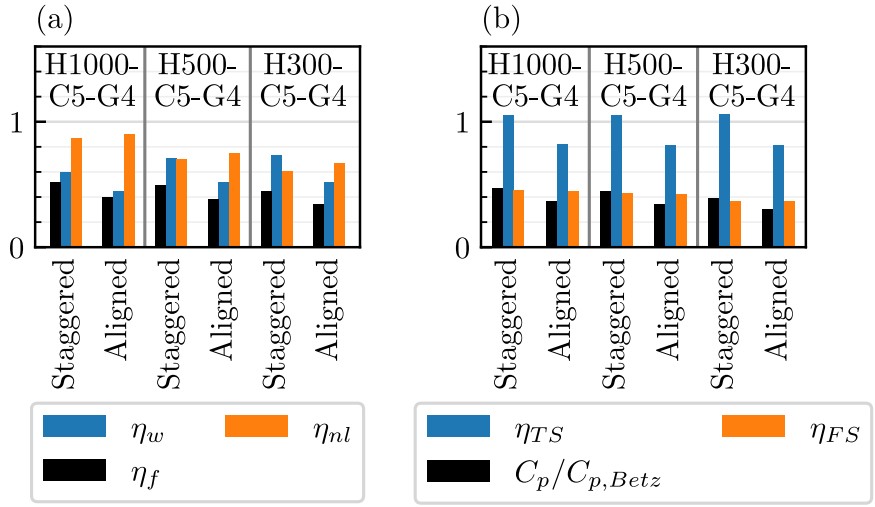

**Figure 16.** Comparison of farm performance for staggered and aligned turbine layouts in H1000-C5-G4, H500-C5-G4 and H300-C5-G4 atmospheric conditions using **(a)** wake efficiency $\eta_w$ and non-local efficiency $\eta_{nl}$ and **(b)** turbine-scale efficiency $\eta_{TS}$ and farm-scale efficiency $\eta_{FS}$.

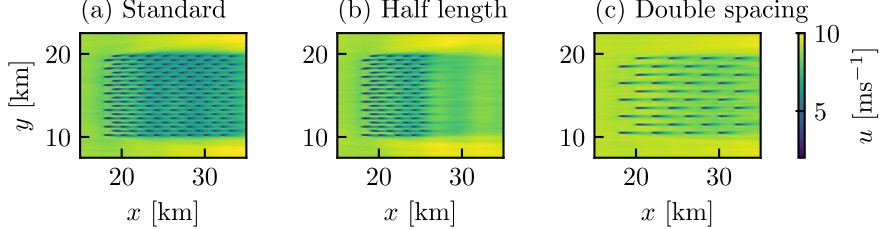

**Figure 17.** Time-averaged $u$ velocity contours at the turbine hub height for **(a)** 'standard' turbine layout, **(b)** 'half length' layout and **(c)** 'double spacing' layout.

It is worth noting that the staggered turbine layout with a $5D \times 5D$ spacing consistently gives $\eta_{TS}$ of approximately 1.05 (see

Fig. 15 and 18), meaning that, on the turbine-scale, the turbines are more efficient at extracting power than isolated turbines. This is presumably due to the 'local blockage' effect caused by neighbouring turbines (Ouro and Nishino, 2021; Nishino and Draper, 2015). It is important to note that $\eta_{TS} = 1$ does not mean the maximum possible performance at the turbine-scale. It means the performance, at the turbine-scale, is equivalent to an isolated turbine that experiences the farm-average wind speed. The performance of an isolated turbine can be exceeded slightly due to local flow confinement effects. When the turbine

spacing was doubled $\eta_{TS}$ was reduced to 0.975 (Fig. 18(b)). This shows that the close turbine spacing caused $\eta_{TS}$ to be greater





than 1. Note that while a close lateral turbine spacing can increase $\eta_{TS}$ slightly above 1, it also reduces $\eta_{FS}$ thereby reducing the overall farm efficiency $C_p/C_{p,Betz}$.

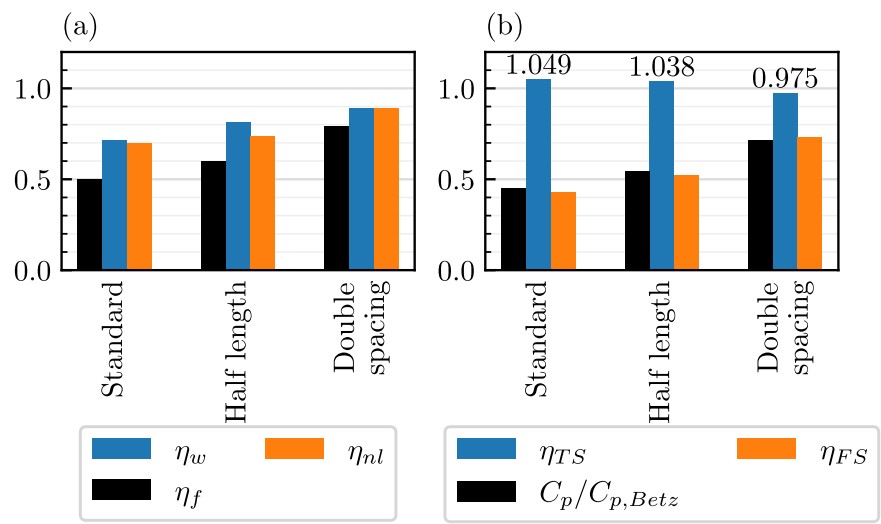

**Figure 18.** Comparison of farm performance for 'standard', 'half-length' and 'double spacing' turbine layouts under H500-C5-G4 atmospheric conditions using **(a)** wake efficiency $\eta_w$ and non-local efficiency $\eta_{nl}$ and **(b)** turbine-scale efficiency $\eta_{TS}$ and farm-scale efficiency $\eta_{FS}$.

### 4.4 Analytical wind farm model

In this section we assess the accuracy of an analytical wind farm model (Kirby et al., 2023b) to predict $\eta_{FS}$. This analytical
model predicts the farm-scale flows only and not the turbine-wake interactions. Hence, here we only compare the predictions of the farm-scale efficiency $\eta_{FS}$ and not the turbine-scale efficiency $\eta_{TS}$. A summary of this farm model is shown in Fig. (19). Essentially, rather than using $\zeta_{LES}$ to calculate $\eta_{FS}$, here we predict $\eta_{FS}$ using the expression

$$\zeta = 1.18 + \frac{\frac{2.18}{C_{f0}}\frac{H_F}{L}}{1 - \frac{\tau_{t0}}{\tau_{w0}}} \tag{20}$$

where $L$ is the streamwise farm length and $\tau_{t0}$ is the undisturbed shear stress at a height $H_F$ in the hub-height wind direction.
Using this approach, we can predict $\eta_{FS}$ instantly using only the undisturbed atmospheric conditions. Figure 20(a) shows the predictions of $\eta_{FS}$ for three different atmospheric conditions. The model correctly predicts the effect of capping inversion layer height on $\eta_{FS}$. It is important to note that this model (Eq. (20)) currently only considers the impact of capping inversion height, and not capping inversion strength or free-atmosphere stratification. Figure 20(b) shows the distribution of percentage errors when predicting $\eta_{FS}$ for 29 atmospheric states (we excluded the lowest capping inversion cases 'H150' where the



---

**Input parameters**

| | |
|---|---|
| Turbine/farm design | $C_T'$, $\lambda$, $L$, $H_F$ |
| Environmental | $C_{f0}$, $\dfrac{\tau_{t0}}{\tau_{w0}}$ |

Step 1. Obtain $\beta$ from $C_T^* \dfrac{\lambda}{C_{f0}} \beta^2 + \beta^2 = 1 + \left[ 1.18 + \dfrac{\frac{2.18}{C_{f0}} \frac{H_F}{L}}{1 - \frac{\tau_{t0}}{\tau_{w0}}} \right] (1 - \beta)$

n.b. $C_T^*$ is calculated using Eq. (4) and is then multiplied by $1/N^2$ to account for LES resolution effects

Step 2. Obtain $\eta_{FS}$ from $\eta_{FS} = \beta^3$

---

**Figure 19.** Input parameters and procedure used to calculate $\eta_{FS}$ from the analytical model (Kirby et al., 2023b)

.

capping inversion was below $H_F$). The model gives good predictions of $\eta_{FS}$ with a mean absolute percentage error of 5.68%. It is likely that the impact of free atmospheric stratification, not considered in the current model, causes some spread and contributes to this error. There is a slight bias for underpredicting $\eta_{FS}$ but the median percentage error is below 5%. The prediction accuracy for the smaller farm and greater turbine spacing cases are also summarised in table 5, showing that this first-order model gives satisfactory predictions for these cases as well.

**Table 5.** LES and analytical model predictions of $\eta_{FS}$ for 'standard', 'half length' and 'double spacing' layouts under H500-C5-G4 atmospheric conditions.

| Case | $\eta_{FS}$ (LES) | $\eta_{FS}$ (Analytical model) | Percentage error (%) |
|---|---|---|---|
| Standard | 0.427 | 0.413 | -3.59% |
| Half length | 0.523 | 0.579 | +10.7% |
| Double spacing | 0.733 | 0.744 | +1.47% |





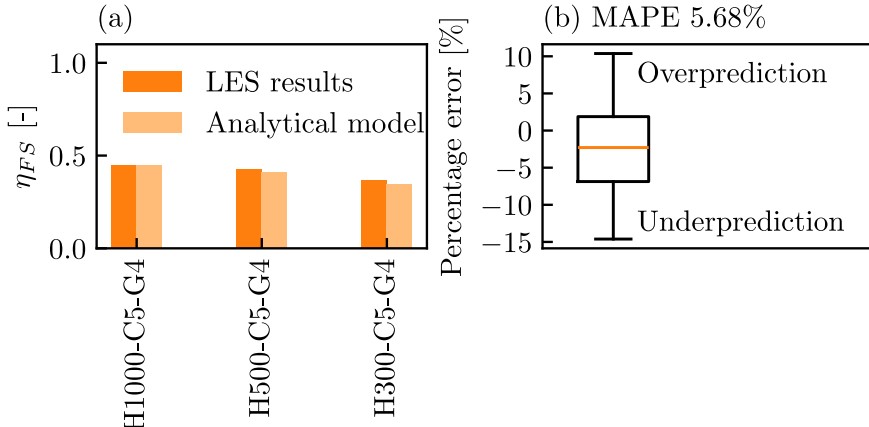

**Figure 20. (a)** Comparison of farm-scale efficiency $\eta_{FS}$ obtained from LES and analytical predictions (Kirby et al., 2023b) for 3 cases and **(b)** box plot showing the distribution of prediction percentage errors for 29 cases.

## 5 Discussion

Power losses for downstream rows of turbines in a wind farm (relative to the first row) are commonly attributed to wake effects. For large farms, the downstream power losses are also affected by the atmospheric response, but it has been a challenge to model this effect accurately. In this study our LES results showed that, for a large staggered array of 160 turbines, the downstream power degradation was not due to turbine-wake interactions but entirely due to the farm-atmosphere interaction. It is worth noting that Porté-Agel et al. (2013) used LES to show that turbine wake effects could reduce farm power by up to 35% by simulating different wind directions. However, this power loss was calculated relative to the optimal wind direction and not the power of the front row turbines. Furthermore, the wind farm simulated by Porté-Agel et al. (2013) was less than 20% the size of the 'standard' farm considered in this study. A recent study by Kirby et al. (2022) suggests that the relative importance of power losses due to turbine-wake interactions decreases with increasing farm size. The present study further supports the argument that the turbine-wake interactions do not play a leading role in power losses for large offshore wind farms.

Our LES results also showed that there was only a weak relationship between the farm blockage loss and the overall farm efficiency. This was due to the strong negative correlation between the blockage loss (represented by $\eta_{nl}$) and the wake loss (represented by $\eta_w$). This suggests that farm blockage, to a first order, acts to redistribute power across the farm rather than reduce the farm power. This has also been observed in the LES performed by Lanzilao and Meyers (2022) and Stipa et al. (2023). The different stratifications changed the farm blockage but the overall farm efficiency changed only slightly.

The turbine-scale efficiency $\eta_{TS}$ and farm-scale efficiency $\eta_{FS}$ are useful new metrics for understanding wind farm performance. They allow the impacts of turbine layout and farm-atmosphere interaction to be assessed separately. The farm-scale efficiency $\eta_{FS}$ is insensitive to the turbine layout and so the losses due to the atmospheric interaction can be assessed even before the turbine layout is decided. It is worth noting that, although actuator discs (or ideal turbines) were considered in this



study, the new metrics $\eta_{FS}$ and $\eta_{TS}$ can be applied to wind farms with real (non-ideal) turbines as well. When the turbines are non-ideal, the power loss due to turbine design (relative to the power of ideal actuator discs) will reduce $\eta_{TS}$ (as $C_p$ in Eq. (13) decreases) but not $\eta_{FS}$.

For all turbine layouts and atmospheric conditions considered in this study, $\eta_{FS}$ was lower than $\eta_{TS}$. This means that more power is lost due to the farm-atmosphere interaction than due to turbine-wake interactions. This was true even for the large farms with an aligned layout and close turbine spacing, where $\eta_{TS}$ was about 0.8 whilst $\eta_{FS}$ was less than 0.5. All staggered cases gave $\eta_{TS}$ close to 1, suggesting no negative turbine-wake interactions. These results suggest the importance of focusing more on the modelling of $\eta_{FS}$ (or the modelling of wind extractability factor $\zeta$) in future studies of large wind farms.

In this study the assumption of 'two-scale separation' was shown to be valid for large finite wind farms. This means that the modelling of large wind farms could be split into the modelling of turbine-wake interactions and the modelling of farm-atmosphere interactions, as suggested originally by Nishino and Dunstan (2020). It should be noted, however, that the present LES results are still for an idealised wind farm situation, i.e., quasi-steady situation with a horizontally homogeneous atmosphere. We found that the wind extractability factor $\zeta$ was insensitive to the internal/turbine-scale flow conditions (i.e., turbine layout and spacing). However, there may still be ways to manipulate the turbine-scale flows to increase $\zeta$, and hence the overall farm efficiency. One way could be to introduce some medium-scale unsteadiness by varying turbine operating conditions in time and thereby increasing momentum entrainment into the farm (e.g., Goit and Meyers, 2015).

Another limitation of the present study is that we relied on a single LES dataset. In this study we focused mostly on a large and relatively dense wind farm. To test the applicability to a wider range of wind farm situations, future work can apply the new concepts of $\eta_{TS}$ and $\eta_{FS}$ to various wind farm LES with different flow conditions (e.g., Baas et al., 2023). Future work could also work on validating the proposed models against wind farm SCADA data. The environmental input parameters ($C_{f0}$ and $\tau_{t0}/\tau_{w0}$) could be calculated using ERA5 data. Data on the surface shear stress and boundary layer height from ERA5 could be used to estimate the shear stress ratio $\tau_{t0}/\tau_{w0}$.

Future work should also focus on improving the analytical model of the momentum availability factor $M$ and the wind extractability factor $\zeta$ (Kirby et al., 2023b). One improvement could be to explicitly model the impact of gravity waves on the farm pressure field (e.g., Smith, 2024). It will also be useful to improve the modelling of turbine-scale flows. Kirby et al. (2023a) developed a statistical model to predict $C_T^*$ as a function of turbine layout for a fixed $C_T'$ value of 1.33. Future work can extend this model to other turbine operating conditions.

## 6   Conclusions

In this study we analysed a large LES suite of wind farms in CNBLs. For all 38 simulation cases with the same staggered turbine layout, the overall farm efficiency $\eta_f$ was not well correlated with the wake efficiency $\eta_w$ (often referred to as normalised power) or the non-local efficiency $\eta_{nl}$ (i.e., farm blockage effects). Identical turbine layouts with different atmospheric stratifications (above the turbines) were found to give significantly different $\eta_w$ values, which could not be explained by changes in



'effective' turbine layout (due to changes in local wind direction) or changes in the rate of wake recovery. These results suggest that turbine-wake interactions do not play a leading role in downstream power losses in large wind farms.

The assumption of two-scale separation (Nishino and Dunstan, 2020) was validated in this study, using finite-size wind farm LES for the first time. The 'internal' parameter $C_T^*$ was found to be insensitive to 'external' atmospheric conditions, whereas the 'external' parameter $\zeta$ was shown to be insensitive to the turbine layout. This means that the assumption of two-scale separation is valid for large offshore wind farms, at least under the ideal 'quasi-steady' situation considered in this study.

Building upon the two-scale momentum theory, we have proposed new metrics of wind farm efficiency. The turbine-scale efficiency $\eta_{TS}$ represents power losses due to internal turbine-wake interactions. The farm-scale efficiency $\eta_{FS}$ reflects the losses due to the farm-atmosphere interaction. As can be expected from the validity of the two-scale separation, $\eta_{TS}$ and $\eta_{FS}$ are useful concepts for understanding the aerodynamic performance of wind farms. For all turbine layouts simulated, $\eta_{FS}$ was found to be much lower than $\eta_{TS}$. This means that farm-scale flows have a greater impact on the overall farm efficiency than turbine-scale flows. The analytical model developed recently by Kirby et al. (2023b) was shown to predict $\eta_{FS}$ with an average error of 5.68% from the LES results. Further developments in the modelling of farm-scale efficiency $\eta_{FS}$ will be crucial in future studies of large wind farms.

*Code and data availability.* The code to reproduce the results and figures is available in the github repository https://github.com/AndrewKirby2/LES_CNBL_analysis. The LES data is available at the KU Leuven RDR dataset https://doi.org/10.48804/L45LTT.

*Author contributions.* T.N. derived the theory. L.L. performed the simulations. A.K. analysed the data from the simulations. A.K. and T.N. drafted the manuscript with guidance from L.L., T.D.D, and J.M. Funding was acquired by T.N., T.D.D. and J.M.

*Competing interests.* At least one of the (co-)authors is a member of the editorial board of *Wind Energy Science*.

*Acknowledgements.* A.K. acknowledges the NERC-Oxford Doctoral Training Partnership in Environmental Research (NE/S007474/1) for funding and training. L.L. and J.M. acknowledge support from the Research Foundation Flanders (FWO, Grant No. G0B1518N), from the project FREEWIND, funded by the Energy Transition Fund of the Belgian Federal Public Service for Economy, SMEs, and Energy (FOD Economie, K.M.O., Middenstand en Energie) and from the European Union Horizon Europe Framework programme (HORIZON-CL5-2021-D3-03-04) under grant agreement no. 101084205. The computational resources and services in this work were provided by the VSC (Flemish Supercomputer Center), funded by the Research Foundation Flanders (FWO) and the Flemish Government department EWI.





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
