# Peer review of "Turbine- and farm-scale power losses in wind farms: an alternative to wake and farm blockage losses"

_Wind Energy Science, 2024_

## Referee Comment (RC1)

**Review of "Turbine- and farm-scale power losses in wind farms: an alternative to wake and farm blockage losses" by A. Kirby, T. Nishino, L. Lanzilao, T. Dunstan, J. Meyers**

This paper analyses large-eddy simulation (LES) results of finite-sized large wind farms subjected to a conventionally neutral boundary layer flow with varying capping inversion heights, strengths and lapse rates. Perfectly staggered and aligned configurations are studied, along with half-length and twice-spaced configurations. The LES results are analysed in the context of the two-scale momentum theory developed previously by the authors. Conventional definitions of farm efficiency, wake efficiency and non-local efficiency are all found to be a function of the atmosphere-farm scale interactions as well as of turbine wake-farm interactions. Alternate metrics of farm efficiency are proposed, namely farm-scale efficiency and turbine-scale efficiency. These two metrics are found to be sensitive to either the atmosphere-farm scale interactions alone or to the turbine wake-farm scale interactions alone, but not to both. The product of these two metrics gives the combined effect of atmosphere-farm and farm-turbine wake scale interactions. The LES results show that the power degradation in most of the wind farms studied is primarily because of atmosphere-farm scale interactions and are characterized by small values of the farm-scale efficiency. An empirical model for the farm-scale efficiency is shown to agree with the LES results with reasonable accuracy.

Overall, this is a well-written paper with several novel contributions. The first is the verification that the internal thrust coefficient is a function of turbine array properties alone while the farm availability factor is a function of the atmospheric conditions alone. Second is the novel metrics that clearly respond only to the farm-scale properties or to the atmospheric conditions alone. The last is the analytical model for farm-scale efficiency.

The paper can be improved by providing some more clarity on the LES setup, the algorithm for calculating the turbine-scale and farm-scale efficiencies, and a few intermediate quantities (i.e. quantities that are not directly compared between model and LES, but are needed as a step towards calculating quantities that are compared, e.g. the efficiencies). Please see detailed comments below.

Major Points:

1. Section 3 describes the LES setup very briefly and refers to Lanzilao & Meyers [JFM, v. 979, 2024] for more details. However, there are a few additional details that should be part of this paper itself to make it self-contained. For example, please mention the surface roughness, Coriolis frequency, the driving force (presumably it is a geostrophic wind) and the upstream fetch. Also mention what are the additional 5 simulations performed here.

2. Lines 170 – 180: Fig. 5 plots the wake efficiency against the farm-averaged yaw angle and shows that there is a weak correlation between them. I think it is inappropriate to take an average of the yaw angles across all turbines in a wind farm. This is because all turbines do not yaw in the same direction, i.e. some yaw clockwise and others yaw anticlockwise, as seen in Fig. 7 of Lanzilao & Meyers [JFM, v. 979, 2024]. Thus, farm-averaged power and farm-averaged turbine yaw angles are likely never going to be correlated. Perhaps it would be better to check some measure of power of each turbine against the individual yaw angles across all the LES cases (no. of data points would be 38 cases times the number of turbines in each case) to arrive at a conclusion regarding whether effective turbine layout is correlated with the wind farm performance.

3. Lines 195 – 200: Figs. 8 and 9(b) show that a lower wake efficiency is obtained for higher $k^*$ values. Is the initial wake width ($\varepsilon$) almost the same across the turbines? It is possible that between two wind farms, the wake growth rate ($k^*$) is larger but the total wake width ($k^*x + \varepsilon$) is actually smaller, and hence the wake efficiency is smaller. Do the authors ensure that this does not happen in their LES results?

4. In the algorithm shown in Fig. 13, $\beta$ can be calculated directly from the LES (from velocities $U_F$ and $U_{F0}$). This is used to calculate $M_{LES}$ and then $\zeta_{LES}$. Then another $\beta$ is calculated in Step 3. The existence of two values of $\beta$ is confusing. Is an iterative procedure used, i.e. Steps 1, 2, 3 are repeated until convergence? If not, how different are the values of $\beta$ and $\beta_{LES}$? What is the meaning of two different $\beta$ values? Why not use $\beta_{LES}$ directly in Step 4?

5. Lines 280 – 285: $C_T^*$ values shown in Fig. 11 are not 0.974. What is the justification for using this value for $C_T^*$ in Section 4.3? It does not appear to be adjusted upwards when compared to Fig. 11.

6. Lines 275 – 285: The multiplication by the correction factor N or its powers following Shapiro et al. (2019) seems to be an ad-hoc fix. Are the results of the analytical model sensitive to this ad-hoc fix? I wonder if it is possible to conduct one simulation where these corrections are incorporated and check whether an ad-hoc fix is no longer needed?

Minor Points:

1. Section 2: It would help to know under what conditions (if any), $C_{P, Nishino}$ reduces to $C_P$, i.e. Eq. (6) reduces to eq. (7).

2. Lines 215 – 225: The last paragraph on pg. 12 and first paragraph on pg. 13 refer to Eqs. (11), and (12a), (12b), (12c). However, these equations are written after the text, which is usually not done. Please reorder the text and the equations and reword appropriately.

3. Are the intermediate quantities, such as $T_i$, $U_F$, $U_{F0}$, $C^*_{T, LES}$, needed to compute $M_{LES}$ and $\beta_{LES}$, provided in the dataset? It would be very helpful for other researchers to have access to these quantities for all the LES cases.

4. In Eq. (20), is $\tau_{t0}/\tau_{w0}$ obtained from the precursor LES? That seems to be the only parameter that responds to the atmospheric conditions and is the key that leads to different wake efficiencies. It would be instructive to show this value for the three cases in Fig. 20.

---

## Referee Comment (RC2)

**Referee's comments to wes-2024-79**

This work validates the two-scale farm modeling framework by Nishino using a large set of LES simulations. The topic is relevant, especially as wind farms are growing in size and start interacting more with the top of the boundary layer. Results are interesting and well explained.

The only main point I would like the authors to revisit it the interpretation of $\eta_{TS}$ and $\eta_{FS}$. The farm-scale efficiency, $\eta_{FS}$, is basically the average efficiency of the turbines in the farm normalized by the maximum efficiency of an isolated turbine derived using Nishino's approach.

According to Nishino's model, the farm is modeled as a distributed sink of momentum, like the equivalent roughness model proposed by Frandsen. It means that the $\eta_{FS}$ will encompass in an average and distributed sense the effect of the thrust of individual machines. This does include effects that are very much "local", like wakes, but in a spatially-averaged sense and thus unaware of the specific layout.

The turbine-scale efficiency, $\eta_{TS}$, represent *de facto* a correction on top of Nishino's model to account for specific farm layouts. The fact that it is >1 sometimes it is not surprising in my opinion, being the farm-scale counterpart, $\eta_{FS}$, based on an average layout. Naturally, some layouts will have $\eta_{TS}$>1, some $\eta_{TS}$<1.

From the discussion in the paper, it sounds like $\eta_{FS}$ should encompass "atmosphere-to-farm interaction" while $\eta_{TS}$ only local effect. I find this distinction a bit misleading because:

- $\eta_{FS}$ could be less than 1 even in a non-atmospheric flow as a consequence of wakes
- $\eta_{TS}$ appears at first glance to include all wake effects, while instead it is only a correction for the departure of the layout from the spatially-averaged one
- $\eta_{TS} > 1$ is interpreted as a "turbine performing better than an isolated one but in a farm" which is a quite a contradicting statement.

Long story short, I feel that $\eta_{TS}$ was given too much physical importance while it is simply a correction factor for the equivalent-roughness model. Further interpretations like "local wakes", "flow confinements" seem not evidenced-based.

I think it should be made clearer in introduction, discussion, and conclusion that the scale separation is more a convenient modeling tool rather than the result of different physics playing out. See e.g. discussion in Stevens et al. [1].

I think that after rephrasing the discussion based on these suggestions and the one below, the paper can be accepted for publication.

**Specific comments**

L22: "Measurements" sound too generic, the cited papers refer to operational turbine data. Indeed, there is vast literature of wake observation through remote sensing (e.g. [2,3,4]) that it is worth mentioning.

L31: the fact that there are velocities reduction within the farm "in addition" to wake is quite philosophical. Apart from pressure-induced effects (like blockage, channelizations, speedups), one could argue that all the momentum deficit in the ABL is the result of superposed wakes. Also, the internal boundary layer growth can be seen as merging and vertically expanding wakes. It should be made clearer that the distinction between the "wakes" and the "farm effects" is merely based on the spatio-temporal scales considered and not due to intrinsically different physics.

L47: I suggest revisiting the word "validate" when referring to the two-scale hypothesis. "Assess", "test" sound more appropriate and less definitive.

L63: $\tau_w$ may have not been defined.

Eq. 3: Nishino and Dunstan also have a $\sigma_1$ factor in their Cp equation, please justify $\sigma_1 \sim 1$ used here.

L82: "upper limit" with respect to which independent variable? Is the maximum Ct attainable by changing the induction of the turbines (like Betz's theory)?

L85: $C_T'$ should have an $i$ index but it does not. If as stated later it is assumed constant, it is a good point to state it (e.g. "the i-index is dropped because we assume […]")

Eq. 4: please explain $\alpha$ right after the equation.

L91: is the thrust or thrust coefficient that needs to be uniform across the farm?

Eq 5: please define explicitly $C_p$. Is it the average power over the farm divided by an available kinetic energy? Is it the average of the individual $C_p$? Or something else?

Fig. 13: why do you use a $\beta_{LES}$ (presumably equal to the velocity ratio $U_F/U_{F0}$) and then a $\beta$ from Eq. 1 again? I understand that the first two steps are needed to estimate the $\zeta$ which is the only unknown of the model. However, there should be information on, for instance, how close the $\beta_{LES}$ is from the $\beta$, which can be an indication of the physical soundness of Nishino's model based on control volume analysis vs LES.

Fig. 14: The interpretation of these results it is not very compelling. Here we are comparing farms with the same layout, same capping inversion heights and free atmosphere lapse rate, but different capping inversion strengths (i.e. different blockages and momentum entrainment). These are my take aways:

- When using $\eta_w$, $\eta_{nl}$, results are not really meaningful because they are based on the assumption that the first row is representative of isolated turbine power, which breaks down in case of blockage.
- $\eta_{FS}$ is capturing most of the energy losses due to blockage and also wakes (which are local effects), but in an average sense and thus not connected to the farm layout. In other words, $C_{p.Nishino}$ is the efficiency of the farm (including wakes!) but for all possible layouts. Calling this "farm-atmosphere interaction losses" is misleading. $C_{p.Nishino}$ would be less than 1 even

in a non-stratified, uniform inflow, just because of wakes. The fact that $C_{p.Nishino} \sim C_p/C_{p,Betz}$ simply means that the layout considered happens to have losses similar to the average layout adopted by Nishino.

- $\eta_{TS}$ is only a small correction that accounts for local layout effects not considered in the global Nishino model. I don't agree that this means that the "turbines perform better than if they were isolated" It simply means to me that this particular layout has slightly lower losses than the average layout considered by Nishino.

Fig 16.: I would make this figure bigger, as it is arguably the most important. It shows that the $\eta_{TS}$ capture changes in the layout (which is evident) and should show that $\eta_{FS}$ should track the changes in efficiency due to stability. The latter is not very clear since values are similar across different capping inversion heights. I suggest adding the number of not of each bar.

L 326: The conclusion that flow confinement is causing the $\eta_{TS} > 1$ are not supported by specific evidence here. The local-scale efficiency larger than 1 simply means that the turbines do better than those in an average layout. The average layout can be interpreted as an infinitely large fetch of rough elements exerting the same thrust as the turbines over a unit area. $\eta_{TS}$ will be greater or lower than one for every departure form this idealized average layout. If it is flow confinement or other effects, it was not shown.

Section 4.4.: the error analysis of the analytical model could be made more comprehensive. A linear regression between all the farm efficiencies from LES and model with error metrics (e.g., $R^2$) should be shown instead of only the overall error (Fig. 20b)

**References**

[1] Stevens, R. J., Gayme, D. F., & Meneveau, C. (2015). Coupled wake boundary layer model of wind-farms. Journal of renewable and sustainable energy, 7(2).

[2] Hirth, B. D., & Schroeder, J. L. (2013). Documenting wind speed and power deficits behind a utility-scale wind turbine. Journal of Applied Meteorology and Climatology, 52(1), 39-46.

[3] Hasager, C. B., Vincent, P., Badger, J., Badger, M., Di Bella, A., Peña, A., ... & Volker, P. J. (2015). Using Satellite SAR to Characterize the Wind Flow around Offshore Wind Farms, Energies, 8, 5413–5439.

[4] Zhan, L., Letizia, S., & Valerio Iungo, G. (2020). LiDAR measurements for an onshore wind farm: Wake variability for different incoming wind speeds and atmospheric stability regimes. Wind Energy, 23(3), 501-527.

---

## Author Comment (AC1)

**[wes-2024-79] Authors' response to Referee #1**

We thank the referee for providing very positive and constructive comments on our manuscript. In the following we provide our response to each of the points raised by the referee (with changes to the manuscript highlighted in red).

Major Points:

1. Section 3 describes the LES setup very briefly and refers to Lanzilao & Meyers [JFM, v. 979, 2024] for more details. However, there are a few additional details that should be part of this paper itself to make it self-contained. For example, please mention the surface roughness, Coriolis frequency, the driving force (presumably it is a geostrophic wind) and the upstream fetch. Also mention what are the additional 5 simulations performed here.

A. We have added the following details in Section 3 of our revised manuscript:

*The simulations are performed with SP-Wind, an in-house LES code developed at KU Leuven (Allaerts and Meyers 2017, Lanzilao and Meyers 2023a). The streamwise (x) and spanwise (y) directions are discretized with a Fourier pseudo-spectral method. For the vertical dimension (z), an energy-preserving fourth-order finite difference scheme is adopted (Verstappen and Veldman 2003). The effects of subgrid-scale motions on the resolved flow are taken into account with the stability-dependent Smagorinsky model proposed by Stevens, Moeng and Sullivan (2000) with Smagorinsky coefficient set to Cs = 0.14. The constant Cs is damped near the wall by using the damping function proposed by Mason and Thomson (1992).*

*To break the streamwise periodicity and impose an inflow condition, we use the wave-free fringe region technique (Lanzilao & Meyers 2023a). At the top of the domain, a rigid-lid condition is used, which implies zero shear stress and vertical velocity and a fixed potential temperature. To minimize gravity-wave reflection, we adopt a Rayleigh damping layer in the upper part of the domain.*

*In this study we fix the geostrophic wind to 10 m s$^{-1}$, which is in line with previous studies (Abkar and Porté-Agel 2013; Wu and Porté-Agel 2017; Allaerts and Meyers 2017, 2018; Lanzilao and Meyers 2022). This value is also chosen so that all turbines operate below their rated wind speed, justifying the use of constant thrust coefficient noted earlier. Finally, we fix the Coriolis frequency to $f_c$ = 1.14 × 10$^{-4}$ s$^{-1}$, and the surface roughness to $z_0$ = 1 × 10$^{-4}$ m for all simulations.*

2. Lines 170 – 180: Fig. 5 plots the wake efficiency against the farm-averaged yaw angle and shows that there is a weak correlation between them. I think it is inappropriate to take an average of the yaw angles across all turbines in a wind farm. This is because all turbines do not yaw in the same direction, i.e. some yaw clockwise and others yaw anticlockwise, as seen in Fig. 7 of Lanzilao & Meyers [JFM, v. 979, 2024]. Thus, farm-averaged power and farm-averaged turbine yaw angles are likely never going to be correlated. Perhaps it would be better to check some measure of power of each turbine against the individual yaw angles across all the LES cases (no. of data points would be 38 cases times the number of turbines in each case) to arrive at a conclusion regarding whether effective turbine layout is correlated with the wind farm performance.

A. We agree with the referee that all turbines do not yaw in the same direction and therefore it would be inappropriate to take an average of the yaw angles (e.g., positive and negative yaw angles would be cancelled out). However, what we have plotted in Fig. 5 is the farm-averaged "magnitude"

of the yaw angles, and therefore we think that this does give a good indication as to the degree of turbine yawing within the farm.

3. Lines 195 – 200: Figs. 8 and 9(b) show that a lower wake efficiency is obtained for higher k* values. Is the initial wake width (ε) almost the same across the turbines? It is possible that between two wind farms, the wake growth rate (k*) is larger but the total wake width (k*x + ε) is actually smaller, and hence the wake efficiency is smaller. Do the authors ensure that this does not happen in their LES results?

A. We have confirmed that the total wake width also correlates negatively with the wake efficiency (in a similar manner to how the wake growth rate k* does). We have added a new figure 9(c) to show this trend, and we have also added the following sentence:

*This trend can also be confirmed from the negative correlation between $\eta_w$ and the farm-averaged turbine wake width (at 10D downstream of each disc) shown in Fig. 9(c).*

4. In the algorithm shown in Fig. 13, β can be calculated directly from the LES (from velocities $U_F$ and $U_{F0}$). This is used to calculate $M_{LES}$ and then $\zeta_{LES}$. Then another β is calculated in Step 3. The existence of two values of β is confusing. Is an iterative procedure used, i.e. Steps 1, 2, 3 are repeated until convergence? If not, how different are the values of β and $\beta_{LES}$? What is the meaning of two different β values? Why not use $\beta_{LES}$ directly in Step 4?

A. The aim of Step 3 is obtain $\beta$ for the "near-ideal" (hypothetical) wind farm subjected to a given $\zeta_{LES}$. Therefore, the value of $\beta$ (obtained from Step 3) is different from $\beta_{LES}$, and the value of $\beta$ (not $\beta_{LES}$) should be used in Step 4 to calculate the farm-scale efficiency (which is the efficiency of the "near-ideal" farm, not the actual farm simulated in the LES). To make this point clearer, we have added the following sentences to the caption of Fig. 13:

*Note that $C_T^*$ required in Step 3 is not $C_{T,LES}^*$ in Fig. 11 but the theoretical $C_T^*$ given by Eq. (4). This is because the aim here is to obtain $\beta$ for the 'near-ideal' (hypothetical) wind farm subjected to a given wind extractability factor $\zeta_{LES}$ (obtained from LES using Steps 1 and 2).*

These Steps 1 to 3 do not require any iterative procedure, since the equation solved in Step 3 is a quadratic equation for $\beta$, which can be solved analytically.

5. Lines 280 – 285: $C_T^*$ values shown in Fig. 11 are not 0.974. What is the justification for using this value for $C_T^*$ in Section 4.3? It does not appear to be adjusted upwards when compared to Fig. 11.

A. As explained in our response to the previous point, the value of $C_T^*$ used in Step 3 is not $C_{T,LES}^*$ but the theoretical value from Eq. (4). To make this point clearer, we have changed "we used 0.974" to "we used 0.88/N² = 0.974" in our revised manuscript.

6. Lines 275 – 285: The multiplication by the correction factor N or its powers following Shapiro et al. (2019) seems to be an ad-hoc fix. Are the results of the analytical model sensitive to this ad-hoc fix? I wonder if it is possible to conduct one simulation where these corrections are incorporated and check whether an ad-hoc fix is no longer needed?

A. Essentially, the analytical model is not dependent on the correction factor N, since Eq. 20 does not require any information from the wind farm LES results as an input to calculate $\zeta$. The reason why we apply the correction factor N in Step 1 in Fig. 19 is that, in order to make a fair comparison between the analytical model predictions and the farm LES, we need to account for the fact that the actual turbine thrust in the LES is slightly higher than it should be. To make this point clear, we have added the above explanation to the caption of Fig. 19. We agree that it would have been better (less

confusing) if we had adopted the correction factor N in the simulations rather than in this post-processing step, but unfortunately these simulations are computationally expensive and we are unable to run additional simulations in a timely manner.

1.  Section 2: It would help to know under what conditions (if any), $C_{P, Nishino}$ reduces to $C_P$, i.e. Eq. (6) reduces to eq. (7).

A.  Thank you for suggesting this. We have added the following sentence after Eq. (7):

*Note that Eq. (6) reduces to Eq. (7) in two special cases: (i) when $\lambda/C_{f0}$ = 0 and (ii) when $\zeta$ is infinitely large.*

2.  Lines 215 – 225: The last paragraph on pg. 12 and first paragraph on pg. 13 refer to Eqs. (11), and (12a), (12b), (12c). However, these equations are written after the text, which is usually not done. Please reorder the text and the equations and reword appropriately.

A.  Thank you for pointing this out. We have made these changes now.

3.  Are the intermediate quantities, such as $T_i$, $U_F$, $U_{F0}$, $C^*_{T, LES}$, needed to compute $M_{LES}$ and $\beta_{LES}$, provided in the dataset? It would be very helpful for other researchers to have access to these quantities for all the LES cases.

A.  Yes, these data are available in our GitHub repository.

4.  In Eq. (20), is $\tau_{t0}/\tau_{w0}$ obtained from the precursor LES? That seems to be the only parameter that responds to the atmospheric conditions and is the key that leads to different wake efficiencies. It would be instructive to show this value for the three cases in Fig. 20.

A.  We thank the referee for this suggestion, but in our revised manuscript we have removed the original Fig. 20(a) and instead added a new Fig. 20(b) to show the results for all 29 cases instead of the 3 selected cases, following the other referee's suggestion. We believe that this new Fig. 20 is more informative than the original Fig. 20.

---

## Author Comment (AC2)

**[wes-2024-79] Authors' response to Referee #2**

We thank the referee for providing detailed comments on our manuscript. Many of the comments have helped us understand which part of the manuscript should have been explained better. In the following we provide our response to each comment one by one (with changes to the manuscript highlighted in red) but first of all, we would like to highlight 4 major points which the referee seems to have overlooked or misunderstood:

1. The referee states that the two-scale momentum model proposed by Nishino and Dunstan (hereafter referred to as ND20) is "like the equivalent roughness model proposed by Frandsen". However, a key difference is that the equivalent roughness models (also known as 'top-down' models) are for infinitely large wind farms, whereas the ND20 model is for a finite-sized wind farm, i.e., the ND20 model accounts for the effect of wind farm size (as the wind extractability factor $\zeta$ depends on the farm size). To make this point clearer, we have added the following short paragraph on page 3 of our revised manuscript:

*Note that the derivation of Eq. (1) given by Nishino and Dunstan (2020) was for an idealised case where the flow through the farm was assumed to be fully developed. However, they also discussed (in Section 3 of their paper) how the same form of equation could be derived for more general cases, where the net momentum transfer through the side and top surfaces of the farm control volume should also be considered as part of M. See Kirby et al. (2022) for the full expression of M.*

2. The referee mentions "average layout" several times in their comments, and claims that the turbine-scale and farm-scale efficiencies ($\eta_{TS}$ and $\eta_{FS}$) do not have as much "physical importance" as implied in our manuscript, because the ND20 model (from which $\eta_{TS}$ and $\eta_{FS}$ have been derived) is specifically for this "average layout". The question here is whether this "average layout" has any significant meaning in terms of physics, and we believe it does. In our previous LES study (Kirby et al. 2022) we have tested 50 different turbine layouts and showed that most of them give a lower farm-average $C_P$ than the ND20 model prediction (i.e., $\eta_{TS}$ < 1). It is true that some layouts could exceed the ND20 prediction but only slightly (see, e.g., Fig. 12 of Kirby et al. 2022), indicating that the ND20 model does capture important physics. To stress this point, we have added the following sentence in the first paragraph of Section 4.3:

*Kirby et al. (2022) have shown, using LES of flow over a periodic array of actuator discs for 50 different layouts, that the `near-ideal' farm performance predicted by Eq. (6) is a good measure to differentiate the turbine-scale power losses from the farm-scale power losses.*

After the referee's comments, we have realised that the term "ideal farm" (used in our original manuscript) could mean "best farm", which would be misleading in this case, so we have changed "ideal farm" and "ideal power coefficient" to "near-ideal farm" and "near-ideal power coefficient", respectively, in our revised manuscript. We have also added the following sentence on page 4:

*We describe this as 'near-ideal' since this is close to but slightly less than the maximum possible (as shown later).*

3. The referee mentions "$\eta_{FS}$ could be less than 1 even in a non-atmospheric flow as a consequence of wakes". Although the meaning of "non-atmospheric flow" seems a little unclear, we believe that the referee misunderstands our concept of farm-atmosphere interaction here. For example, if we consider a hypothetical scenario where the wind farm is placed in a rectangular channel of height $H_F$ (or consider that a capping inversion layer exists at $z = H_F$ to act as a rigid lid, and the wind is forced

to go through the nominal farm layer of height $H_F$) then we would have $\beta = 1$ and thus $\eta_{FS} = 1$. The point here is that, when each turbine generates its wake, the flow bypassing the turbine must accelerate (due to the conservation of mass at each turbine scale). This means that the generation of turbine wake does not, on its own, cause any reduction of "farm-average" wind speed, and hence, farm-atmosphere interactions are required for the "farm-average" wind speed to decrease (or for the values of $\beta$ and $\eta_{FS}$ to decrease from 1). To explain this point explicitly, we have added the following sentences in the first paragraph of Section 4.3:

*Note that, when each turbine in a wind farm generates its wake, the flow bypassing the turbine locally accelerates due to the conservation of mass (at each turbine scale); hence, we consider that any reduction of farm-average wind speed is caused by external (farm-atmosphere) interactions. This means that the power losses accompanied by a reduction of farm-average wind speed are `farm-scale' power losses (caused by external interactions) and not `turbine-scale' power losses (caused by internal interactions).*

4. The referee mentions that $\eta_{TS} > 1$ is interpreted as "turbine performing better than an isolated one but in a farm". This interpretation is clearly incorrect (or imperfect). The correct interpretation of $\eta_{TS} > 1$ is that the turbines in a farm (with the farm-average wind speed of $U_F$) are performing better than how they would perform when they are isolated and their incoming wind speed is $U_F$. To make this point clearer, we have modified the relevant sentence on page 18 as follows:

*Note that $\eta_{TS}$ is slightly greater than 1, which means that these `clustered' turbines perform slightly better than the isolated ideal turbines (of the same size) that have the same upstream wind speed as the farm-averaged wind speed $U_F$*

Based on the above 4 points, we disagree with the referee's comments that $\eta_{TS}$ is "simply a correction factor for the equivalent-roughness model" and that "the scale separation is more a convenient modeling tool rather than the result of different physics playing out". We hope that the above changes made in our revised manuscript will have resolved the referee's concerns.

**Response to specific comments:**

L22: "Measurements" sound too generic, the cited papers refer to operational turbine data. Indeed, there is vast literature of wake observation through remote sensing (e.g. [2,3,4]) that it is worth mentioning.

A. We thank the referee for suggesting these papers. We have added two of them which we found most relevant to our context.

L31: the fact that there are velocities reduction within the farm "in addition" to wake is quite philosophical. Apart from pressure-induced effects (like blockage, channelizations, speedups), one could argue that all the momentum deficit in the ABL is the result of superposed wakes. Also, the internal boundary layer growth can be seen as merging and vertically expanding wakes. It should be made clearer that the distinction between the "wakes" and the "farm effects" is merely based on the spatio-temporal scales considered and not due to intrinsically different physics.

A. This comment is related to the major point "3" discussed above, and we hope that our answers provided there have explained about our concepts sufficiently.

L47: I suggest revisiting the word "validate" when referring to the two-scale hypothesis. "Assess", "test" sound more appropriate and less definitive.

A.  We believe that it is appropriate to use the word "validate" in some sentences where it does not imply that the two-scale separation assumption has been fully validated. However, following the referee's suggestion, we have changed "validated" to "evaluated" in two sentences where "validated" could sound a little too definitive (on pages 13 and 25 in the revised manuscript).

L63: $\tau_w$ may have not been defined.

A.  Thank you. We have added the definition now.

Eq. 3: Nishino and Dunstan also have a $\sigma_1$ factor in their Cp equation, please justify $\sigma_1 \sim 1$ used here.

A.  Thank you for pointing this out. We have added the following explanation on page 3 where we define the farm-layer height $H_F$ (since this $\sigma_1$ factor is essentially the factor required when $U_{F0}$ differs from $U_{T0}$):

*The exact value of $H_F$ defined originally by Nishino and Dunstan (2020) depends on the undisturbed wind profile, to ensure that $U_{F0}$ matches exactly with the undisturbed wind speed averaged over the turbine swept area, $U_{T0}$; however, as shown later by Kirby et al. (2022) the fixed definition of $H_F = 2.5H_{hub}$ is a good approximation for a wide range of ABL profiles.*

L82: "upper limit" with respect to which independent variable? Is the maximum Ct attainable by changing the induction of the turbines (like Betz's theory)?

A.  This "upper limit" is with respect to the turbine layout (for a fixed value of $C_T'$). We have added "with respect to the turbine layout" in our revised manuscript.

L85: $C_T'$ should have an i index but it does not. If as stated later it is assumed constant, it is a good point to state it (e.g. "the i-index is dropped because we assume [···]")

A.  Yes, $C_T'$ is assumed constant ($C_T'$ = 1.94 in our LES as noted later in Section 3). Now we have stated "assumed to be constant for all turbines in the farm" in the sentence right after Eq. (4).

Eq. 4: please explain $\alpha$ right after the equation.

A.  We have decided to remove $\alpha$ from Eq. (4) in our revised manuscript, since the aim of this equation is to give the (modelled) relationship between $C_T^*$ and $C_T'$.

L91: is the thrust or thrust coefficient that needs to be uniform across the farm?

A.  We have revised the paragraph right before Eq. (5) to explain this point better. It is the turbine resistance coefficient $C_T'$ that is assumed to be uniform across the farm.

Eq 5: please define explicitly $Cp$. Is it the average power over the farm divided by an available kinetic energy? Is it the average of the individual $Cp$? Or something else?

A.  The definition of $C_P$ has already been given at the end of Section 2.1.  It is the "farm-averaged" power coefficient as noted in the sentence right before the equation.

Fig. 13: why do you use a $\beta_{LES}$ (presumably equal to the velocity ratio $U_0/U_{F0}$) and then a $\beta$ from Eq. 1 again? I understand that the first two steps are needed to estimate the $\zeta$ which is the only unknown of the model. However, there should be information on, for instance, how close the $\beta_{LES}$ is from the $\beta$, which can be an indication of the physical soundness of Nishino's model based on control volume analysis vs LES.

A. Here the referee seems to have misunderstood the concept of our analysis summarised in Fig. 13. The aim of our analysis here is obtain $\beta$ for the "near-ideal" (hypothetical) wind farm subjected to a given $\zeta_{LES}$. This means that comparing $\beta$ (obtained from Step 3) against $\beta_{LES}$ will not give an indication of the physical soundness of the theoretical model (because the value of $\beta$ for the "near-ideal" farm should be different from the value of $\beta_{LES}$ for a real farm). To make this point clearer, we have added the following sentences to the caption of Fig. 13:

*Note that $C_T^*$ required in Step 3 is not $C_{T,LES}^*$ in Fig. 11 but the theoretical $C_T^*$ given by Eq. (4). This is because the aim here is to obtain $\beta$ for the 'near-ideal' (hypothetical) wind farm subjected to a given wind extractability factor $\zeta_{LES}$ (obtained from LES using Steps 1 and 2).*

It should also be noted that the soundness of the two-scale approach has already been evaluated in Section 4.2. The focus of Section 4.3 is on its application (rather than its evaluation).

Fig. 14: The interpretation of these results it is not very compelling. Here we are comparing farms with the same layout, same capping inversion heights and free atmosphere lapse rate, but different capping inversion strengths (i.e. different blockages and momentum entrainment). These are my take aways:

•When using $\eta_w$, $\eta_{nl}$, results are not really meaningful because they are based on the assumption that the first row is representative of isolated turbine power, which breaks down in case of blockage.

•$\eta_{TS}$ is capturing most of the energy losses due to blockage and also wakes (which are local effects), but in an average sense and thus not connected to the farm layout. In other words, $C_{P,Nishino}$ is the efficiency of the farm (including wakes!) but for all possible layouts. Calling this "farm-atmosphere interaction losses" is misleading. , $C_{P,Nishino}$ would be less than 1 even in a non-stratified, uniform inflow, just because of wakes. The fact that $C_{P,Nishino} \sim C_P / C_{P,Betz}$ simply means that the layout considered happens to have losses similar to the average layout adopted by Nishino.

•$\eta_{TS}$ is only a small correction that accounts for local layout effects not considered in the global Nishino model. I don't agree that this means that the "turbines perform better than if they were isolated" It simply means to me that this particular layout has slightly lower losses than the average layout considered by Nishino.

A. We believe that our response to the 4 major points (provided at the beginning of this response letter) have sufficiently addressed all these points.

Fig 16.: I would make this figure bigger, as it is arguably the most important. It shows that the $\eta_{TS}$ capture changes in the layout (which is evident) and should show that $\eta_{FS}$ should track the changes in efficiency due to stability. The latter is not very clear since values are similar across different capping inversion heights. I suggest adding the number of not of each bar.

A. We thank the referee for this suggestion. We have made Fig. 16(b) bigger and also added the values of $\eta_{FS}$ (above each orange bar) to show more clearly that $\eta_{FS}$ does change with atmospheric conditions. We have also decided to remove Fig. 16(a) since this figure was not discussed in the main text.

L 326: The conclusion that flow confinement is causing the $\eta_{TS}>1$ are not supported by specific evidence here. The local-scale efficiency larger than 1 simply means that the turbines do better than those in an average layout. The average layout can be interpreted as an infinitely large fetch of rough elements exerting the same thrust as the turbines over a unit area. $\eta_{TS}$ will be greater or

lower than one for every departure form this idealized average layout. If it is flow confinement or other effects, it was not shown.

A. We disagree with the referee's interpretation of $\eta_{TS} > 1$ as we explained through the 4 major points described earlier. However, we agree with the referee that, in this paper, we were unable to provide a clear evidence of flow confinement effects causing $\eta_{TS} > 1$ (although the results for the "double spacing" case shown in Fig. 18, together with the LES results of Ouro and Nishino (2021) cited, suggest that such flow confinement effects are likely the cause of $\eta_{TS} > 1$). We have therefore made some minor changes of wording in our revised manuscript.

Section 4.4.: the error analysis of the analytical model could be made more comprehensive. A linear regression between all the farm efficiencies from LES and model with error metrics (e.g., $R2$) should be shown instead of only the overall error (Fig. 20b)

A. We thank the referee for this suggestion. Now we have added a new Fig. 20b to show the relationship between the LES results and analytical model predictions of $\eta_{FS}$ for all 29 cases, together with the $R^2$ value. We have also removed the original Fig. 20a since this figure was only for 3 selected cases and it was not very informative.

We thank the referee again for all these comments, which have helped us improve the manuscript significantly.

---

## Referee Report (RR1)

**Referee's comments to first revision of wes-2024-79**

**General comments**

Thanks to the authors for making changes to the manuscript. There are however still important aspects that were not addressed as many comments were disregarded.

First, in the manuscript it could be made clearer for the reader who is not familiar with the previous publications where a fully developed flow is assumed ($U_{T,i} = const$) or where the less strict assumption of $C_T' = const$ applies, as now they are kin of mixed. The source of confusion seems to be that ND20 was derived for fully-developed flow (aka, infinite layout), but then they hinted at a more general solution in section 3. Also, Kirby et al. 2020, simulates only infinite farms with actual LES, but then postprocesses the $C_p$ for hypothetical finite-size farm (section 4.3).

Second, the main point, i.e. the physical interpretation of $\eta_{FS}$ and $\eta_{TS}$, was not addressed. The major critical aspects still are:

- $\eta_{TS} = C_p/C_{p,Nishino}$ is the ratio of the "true" power coefficient (i.e. based on LES) and the prediction by ND20 that uses an averaged approach. Mathematically, this simply carries out the modeling error of ND20. It could be applied to every analytical model. The fact that $C_{p,Nishino}$ has a physical meaning for a real layout, other than something like "the efficiency that the farm would have if it was infinite" has not been proven. Calling it the "near-ideal" case in a "best-performing" sense, pointing to the fact the LES simulations of infinite farms (so already pretty idealized) rarely exceeded this value is not rigorous, unless we assume that the 50 LES represent a representative statistical set of all the possible wind farm configurations and atmospheric conditions (which clearly do not as they are infinite). Also, interpreting $\eta_{TS} > 1$ as due to flow confinement is admittedly not proven and it was kept in spite of the previous comments.

- The fact that $\eta_{FS} = C_{p,Nishino}/C_{p,Betz}$ is related "farm-atmosphere interaction" may be misleading. It reminds of calls to mind atmospheric flow features, like Coriolis, stability, mesoscale circulation, blockage, gravity waves, but it could be applied to any neutral flow past an obstacle, not necessarily an "atmospheric" one. Even more questionable are statements like:

  *"In this study our LES results showed that, for a large staggered array of 160 turbines, the downstream power degradation was not due to turbine-wake interaction but entirely due to the farm-atmosphere interaction."*

  Things could be simpler than that: $\eta_{FS}$ mathematically is simply the prediction of efficiency by ND20, with all its limitations. And it does include wakes, because when averaging for instance velocity within the farm layer, wakes do contribute to reduce $U_F$.
  The example of the farm in channel where mass conservation creates speedups that cancel out with wakes is just a very special and realistic case of a pressure-drive flow or, $-\frac{\delta p}{\delta x}$. The momentum deficit in real farms is replenished by reduction in kinetic energy, $U\frac{\partial U}{\partial x}$ if the flow is not fully developed (most cases) and partly turbulence momentum transfer from above and

the sides, $-\frac{\partial \overline{uu_i}}{\partial x_i}$, whereas large scale pressure patterns are minimally changed. Therefore, wakes do contribute to make $\beta < 1$. In other words, saying that $\eta_{FS}$ is not connected to turbine-wake interaction, it is like saying that wakes (and thus thrust) are not considered in ND20, which is a patent contradiction.

Long story short, there it is still not convincing that the $\eta_{TS}$ includes "local effects" and $\eta_{FS}$ quantifies the "farm-atmosphere interaction", which is also an elusive concept. The original idea of the scale separation of ND20 was to isolate on one side of the equations parameters that depend on the layout and wind direction $(C_T^*, \gamma)$, and on the other side the large-scale momentum replenishment from above the farm by the enhanced momentum flux ($M(\zeta)$, and coupled then through $\beta$. This fact was nicely validated in this work. However, relating the prediction ND20, as a whole, to farm-atmosphere interaction and the difference to turbine-wake is not sound. ND20 do include local effects like wake in their modeling.

It is recommended to carefully rethink this definition and possibly remove or mitigate their physical interpretation and reduce the scope of the manuscript as a validation of the two-scale separation hypothesis.

---

## Referee Report (RR2)

**Referee's comments to second revision of wes-2024-79**

Thanks to the authors for making another iteration. The small changes did not address the concerns. However, for the sake of our time, the paper can be accepted after addressing minor revisions, and the editors will acknowledge that some of one Referee's comments were pushed back. The manuscript will be released publicly, and readers will have a chance to formulate their own opinions.

**Comments:**

1. The main point remains not addressed. The Referee understands that the LES simulations (finite or infinite layout) show that the $C_T^*$ is fairly constant for a large set of inflows and that the associated $C_{p,Ninshino}$ is a good first-order approximation of the near-ideal (i.e., an upper limit among all layouts) farm performance. However, the conclusions that (i) $C_p/C_{p,Nishino}$ is a measure of local flow effects and that (ii) $C_p/C_{p,Nishino}$ means "no wake effects" are not agreed upon. Again, there is merely a problem with the narrative. $C_p/C_{p,Nishino}$ is mathematically nothing more than a correction on top of the Nishino model, which is certainly unaware of the layout, but also of changes in velocity profile across the turbine rotors, reliant on 1D momentum theory, valid for infinite layouts, etc. Why would $C_p/C_{p,Nishino}$ be a measure for "local effects" only is not sufficiently supported by results.

   The latest change:

   "In this study our LES results showed that, for a large staggered array of 160 turbines, the downstream power degradation was not due to turbine-wake interactions, i.e., individual turbine wakes (or more specifically, local flow regions having a lower flow speed than the "average" flow speed) were not directly causing the reduction of downstream turbine power (in the sense that how the power of downstream turbines would have been reduced if they had been located in such a locally slower flow region)"

   Has honestly made things even more obscure. If we are still in the realm of fluid mechanics, a lower velocity region does cause a very "directly" a reduction in available power downstream. The last sentence in parentheses sounds very philosophical and not understood at all and should be removed/revised.

2. The addition: "The 'double spacing' case gives an even higher $\eta_{FS}$ because of the low array density, which reduces the total farm thrust and thus the vertical mixing due to turbulence compared to the 'standard' case." sounds confusing. A coarser layout will lead to lower farm-scale efficiency just as the result of reduced wake interactions, which brings the $C_{p,Nishino}$ (which included wakes effects at a farm level) closer to the Betz limit (which is the limit for isolated turbine). The authors themselves say Section 2.2 that $\lambda/C_{f0} \to 0$ implies $C_{p,Nishino} \to C_{p,Betz}$ without need to call out "vertical mixing". Please remove the reference to "vertical mixing" which does not play a direct role here.

---

## Author Response (AR2)

**[wes-2024-79] Reply to editor**

Dear Professor Archer,

Thank you for your comments regarding the reviewer 2's comments. We have thought through these comments very carefully and made some further changes to our manuscript. However, we still do not agree with some of the reviewer's comments, as you will see in our response to the reviewer. We believe that we have sufficiently explained about our opinions (on why we do not agree with some of the reviewer's opinions) in our previous and current responses. We hope our response will be found satisfactory by the reviewer, but please let us know if we need to make any further changes.

**[wes-2024-79] Authors' 2nd response to Referee #2**

We thank the referee for reviewing our revised manuscript. Following their additional comments, we have made some further changes to the manuscript (highlighted in blue).

For the first point, regarding the assumption of $C_T' = $ const., we have added the following sentence after Eq. 4 (page 4):

*Note that the two-scale momentum theory summarised in Section 2.1 is for general cases where the turbine thrust $T_i$ and power $P_i$ may vary across the farm, whereas the analytical model described here is for less general cases where the turbine resistance coefficient $C_T'$ is constant across the farm (such as the LES cases shown later in this paper, where $C_T'$ is fixed at 1.94 for all turbines in the farm).*

With regard to the "main point" of the referee's comments, i.e. "the physical interpretation of $\eta_{FS}$ and $\eta_{TS}$", now we understand that the referee's concern mainly comes from the fact that the 50 LES results reported by Kirby et al. (2022) were only for idealised infinitely large farms. Because of this, we suspect the referee did not think that the key finding of Kirby et al. (2022) for those 50 infinitely large farms (i.e., $C_T*$ may only slightly exceeds the theoretical prediction of ND20) was applicable to finite-sized farms in general. However, as the referee also agreed in their comments, our new LES results shown in the present paper have confirmed the two-scale separation for finite-sized farms; in particular, we have shown that the "internal" thrust coefficient $C_T*$ is insensitive to "external" conditions. This two-scale separation means that the upper limit of $C_T*$ (with respect to the turbine layout, for a given value of $C_T'$) should also be insensitive to "external" conditions, and therefore, the aforementioned finding of Kirby et al. (2022) for infinitely large farms is expected for finite-sized farms as well. This is why we believe that it is reasonable to state that the performance prediction of ND20 is "near-ideal" for finite-sized farms as well.

To better explain about this point, we have added the following sentences after the first paragraph of Section 4.3:

*"It should also be noted that the 50 LES results of Kirby et al. (2022) are for idealised infinitely large wind farms; hence, their findings are not directly applicable to finite-sized farms in general. However, as shown in the previous section, our new LES results indicate that the internal thrust coefficient $C_T*$*

*is insensitive to external conditions. This means that the upper limit of $C_T^*$ (with respect to the turbine layout, for a given value of $C_T'$) should also be insensitive to external conditions, supporting our argument that the 'near-ideal' farm performance predicted by Eq. (6) is a good measure for finite farms as well."*

We also understand that the referee's concern comes from our use of the terms "turbine-wake interaction" (causing the reduction of $\eta_{TS}$) and "farm-atmosphere interaction" (causing the reduction of $\eta_{FS}$). For the former, we would like to clarify what we mean by "interaction" here. When $\eta_{TS} = 1$ (or when we say "there is no power loss due to turbine-wake interaction") we do not mean "there is no turbine wake". There are, of course, turbine wakes in the farm, and we have never ignored their existence. What we mean by $\eta_{TS} = 1$ is that, even though there are turbine wakes in the farm, those individual turbine wakes (or more specifically, local flow regions having a lower flow speed than the "average" flow speed) are not directly causing the reduction of downstream turbine power (in the sense that how the downstream turbine power would be reduced if they were located in such a locally slower flow region). We tried to demonstrate and explain about this as clearly as possible in Section 4.1 of our original manuscript, but now we have also made the following changes in Section 5 (Discussion) and Section 6 (Conclusions):

*"In this study our LES results showed that, for a large staggered array of 160 turbines, the downstream power degradation was not due to turbine-wake interactions, i.e., individual turbine wakes (or more specifically, local flow regions having a lower flow speed than the "average" flow speed) were not directly causing the reduction of downstream turbine power (in the sense that how the power of downstream turbines would have been reduced if they had been located in such a locally slower flow region)."* (Page 23)

*"The present study further supports the argument that farm-scale flow effects could play a leading role in power losses for large offshore wind farms."* (Page 24)

*"These results suggest that farm-scale flow effects could play a leading role in power losses in large wind farms."* (Page 25)

For the term "farm-atmosphere interaction", we agree with the referee that this term would remind the reader of "atmospheric flow features, like Coriolis, stability, mesoscale circulation, blockage, gravity waves". However, we still believe that it is reasonable to use this term to explain about the reduction of $\eta_{FS}$ (or the reduction of farm-average wind speed $\beta$) since these atmospheric flow features are indeed the factors affecting the wind extractability factor $\zeta$, which in turn determines $\beta$ and thus $\eta_{FS}$ (for a given set of "internal" conditions, provided that the two-scale separation is valid). The point here is that $C_{p,Nishino}$ (Eq. 6) has been given as a function of $\zeta$, which, by definition, captures all these atmospheric flow effects (through all mechanisms of momentum transfer between inside and outside the farm, including the turbulent momentum transfer noted by the referee).

However, we admit that our current analytical model of $\zeta$ (Eq. 20) is a highly simplified model and ignores some of such atmospheric flow effects. We have therefore checked our revised manuscript and confirmed that we are not giving a wrong impression that our analytical model (Eq. 20) fully captures the effects of such farm-atmosphere interactions.

We have also either removed or replaced some of "farm-atmosphere interaction" in our manuscript with a more appropriate expression, such as *"vertical mixing due to turbulence"* (page 20).

We have also changed the statement "*the downstream power degradation was not due to turbine-wake interactions but entirely due to the farm-atmosphere interaction*" (in the first paragraph of Section 5) as we agree with the referee that this statement was misleading.

We hope that the above response to the referee, together with the additional changes made to the manuscript, will be found satisfactory by the referee and the editor.  We thank the referee again for all their comments.

---

## Author Response (AR3)

**[wes-2024-79] Authors' 3rd response to Referee #2**

We thank the referee once again for reviewing our manuscript, even though it seems that we were unable to reach an agreement on minor points. We also appreciate that the referee suggests our manuscript be published to allow readers to "have a chance to formulate their own opinions".

Note that since text has only been removed from this revision, there are no track changes in blue in the uploaded version.

This time the referee commented on 2 points, and our responses are as follows:

1. We prefer to avoid repeating our previous answers here, but just briefly, we still argue that $C_p/C_{p,Nishino}$ is a good measure for "turbine-scale power loss" (which is, by our definition, the power loss that is not due to reduction of farm-average wind speed). This is because, as the referee also agrees, $C_{p,Nishino}$ is a good approximation of the performance of "near-ideal" wind farm that has only "farm-scale power loss" (which is, by our definition, the power loss due to reduction of farm-average wind speed). We believe that our results shown in Section 4.1 are clear enough to support these concepts.

   Regarding the latest change we made (in the first paragraph of Section 5), we follow the referee's suggestion and remove

   "*(in the sense that how the power of downstream turbines would have been reduced if they had been located in such a locally slower flow region)*"

   The aim of this added sentence was to briefly summarise the results shown earlier in Section 4.1, but we agree that this sentence, on its own, could be confusing to readers.

2. We follow the referee's suggestion and remove

   "*and thus the vertical mixing due to turbulence*"

   as we agree that this could also be confusing to readers. To explain this properly, we would need to go through our recent work (Kirby et al. 2023, Journal of Fluid Mechanics 976) but we prefer to avoid having a long explanation here.